# Test-Time Multimodal Backdoor Detection by Contrastive Prompting

**Yuwei Niu** [* 1] **Shuo He** [* 2] **Qi Wei** [2] **Zongyu Wu** [3] **Feng Liu** [4] **Lei Feng** [5]

## Abstract

While multimodal contrastive learning methods (e.g., CLIP) can achieve impressive zero-shot classification performance, recent research has revealed that these methods are vulnerable to backdoor attacks. To defend against backdoor attacks on CLIP, existing defense methods focus on either the pre-training stage or the fine-tuning stage, which would unfortunately cause high computational costs due to numerous parameter updates and are not applicable in black-box settings. In this paper, we provide the first attempt at a computationally efficient backdoor detection method to defend against backdoored CLIP in the *inference* stage. We empirically find that the visual representations of backdoored images are *insensitive* to *benign* and *malignant* changes in class description texts. Motivated by this observation, we propose BDetCLIP, a novel test-time backdoor detection method based on contrastive prompting. Specifically, we first prompt a language model (e.g., GPT-4) to produce class-related description texts (benign) and class-perturbed random texts (malignant) by specially designed instructions. Then, the distribution difference in cosine similarity between images and the two types of class description texts can be used as the criterion to detect backdoor samples. Extensive experiments validate that our proposed BDetCLIP is superior to state-of-the-art backdoor detection methods, in terms of both effectiveness and efficiency.

## 1. Introduction

Multimodal contrastive learning methods (e.g., CLIP (Radford et al., 2021)) have shown impressive zero-shot classification performance in various downstream tasks and served as foundation models in various vision-language fields due to their strong ability to effectively align representations from different modalities, such as open-vocabulary object detection (Wu et al., 2023), text-to-image generation (Ramesh et al., 2022), and video understanding (Xu et al., 2021). However, recent research has revealed that a small proportion of backdoor samples poisoned into the pre-training data can cause a backdoored CLIP after the multimodal contrastive pre-training procedure (Carlini & Terzis, 2021; Carlini et al., 2023; Bansal et al., 2023). In the inference stage, a backdoored CLIP would produce tampered image representations for images with a trigger, close to the text representation of the target attack class in zero-shot classification. This exposes a serious threat when deploying CLIP in real-world applications.

To overcome this issue, effective defense methods have been proposed recently, which can be divided into three kinds of defense paradigms, as shown in Figure 1: including (a) robust anti-backdoor contrastive learning in the pre-training stage (Yang et al., 2023b), (b) counteracting the backdoor in a pre-trained CLIP in the fine-tuning stage (Bansal et al., 2023), (c) leveraging trigger inversion techniques to decide if a pre-trained CLIP is backdoored (Sur et al., 2023; Feng et al., 2023a). Overall, these defense methods have a high computational cost due to the need for additional learning or optimization procedures. In contrast, we advocate *test-time* backdoor sample detection (Figure 1(d)), which is a more computationally efficient defense against backdoored CLIP, as there are no parameter updates in the inference stage. Intuitively, it could be feasible to directly adapt existing *unimodal* test-time detection methods (Gao et al., 2019; Guo et al., 2023; Liu et al., 2023) to detect backdoored images in CLIP, since they can differentiate backdoored and clean images generally based on the output consistency in the visual representation space by employing specific image modifications, e.g, corrupting (Liu et al., 2023), amplifying (Guo et al., 2023), and blending (Gao et al., 2019). However, the performance of these unimodal detection methods is suboptimal, because of lacking the utilization of the text modality in CLIP to assist backdoor sample detection. Hence we can expect that better performance could be further achieved if we leverage both image and text modalities simultaneously.

In this paper, we provide the first attempt at a computationally efficient backdoor detection method to defend against

---
*Equal contribution [1]Chongqing University [2]Nanyang Technological University [3]Penn State University [4]University of Melbourne [5]Southeast University. Correspondence to: Lei Feng <fenglei@seu.edu.cn>.

*Proceedings of the 42nd International Conference on Machine Learning*, Vancouver, Canada. PMLR 267, 2025. Copyright 2025 by the author(s).

backdoored CLIP in the *inference* stage. We empirically find that the visual representations of backdoored images are *insensitive* to both *benign* and *malignant* changes of class description texts. Motivated by this observation, we propose BDetCLIP, a novel test-time multimodal backdoor detection method based on contrastive prompting. Specifically, we first prompt the GPT-4 (Achiam et al., 2023) to generate class-related (or class-perturbed random) description texts by specially designed instructions and take them as benign (malignant) class prompts. Then, we calculate the distribution difference in cosine similarity between images and the two types of class prompts, which can be used as a good criterion to detect backdoor samples. We can see that the distribution difference of backdoored images between benign and malignant changes of class prompts is smaller than that of clean images. The potential reason for the insensibility of backdoored images is that their visual representations have less semantic information aligned with class description texts. In this way, we can detect backdoored images in the inference stage of CLIP effectively and efficiently. Extensive experiments validate that our proposed BDetCLIP is superior to state-of-the-art backdoor detection methods, in terms of both effectiveness and efficiency.

Our main contributions can be summarized as follows:

- *A new backdoor detection paradigm for CLIP.* We pioneer test-time backdoor detection for CLIP, which is more resource-efficient than existing defense paradigms.

- *A novel backdoor detection method.* We propose a novel test-time backdoor detection method by contrastive prompting, which detects backdoor samples based on the distribution difference between images regarding the benign and malignant changes of class prompts.

- *Strong experimental results.* Our proposed method achieves superior experimental results on various types of backdoored CLIP compared with state-of-the-art detection methods.

## 2. Background & Preliminaries

### 2.1. Multimodal Contrastive Learning

Multimodal contrastive learning (Radford et al., 2021; Jia et al., 2021) has emerged as a powerful approach for learning shared representations from multiple modalities of data such as text and images. Specifically, we focus on **C**ontrastive **L**anguage **I**mage **P**retraining (CLIP) (Radford et al., 2021) in this paper. Concretely, CLIP consists of a visual encoder denoted by $\mathcal{V}(\cdot)$ (e.g., ResNet (He et al., 2016) and ViT (Dosovitskiy et al., 2020)) and a textual encoder denoted by $\mathcal{T}(\cdot)$ (e.g., Transformer (Vaswani et al., 2017)). The training examples used in CLIP are massive image-text pairs collected on the Internet denoted by

$\mathcal{D}_{\text{Train}} = \{(\boldsymbol{x}_i, \boldsymbol{t}_i)\}_{i=1}^{N}$ where $\boldsymbol{t}_i$ is the caption of the image $\boldsymbol{x}_i$ and $N \simeq 400M$. During the training stage, given a batch of $N_b$ image-text pairs $(\boldsymbol{x}_i, \boldsymbol{t}_i) \subset \mathcal{D}_{\text{Train}}$, the cosine similarity for matched (unmatched) pairs is denoted by $\phi(\boldsymbol{x}_i, \boldsymbol{t}_i) = \cos(\mathcal{V}(\boldsymbol{x}_i), \mathcal{T}(\boldsymbol{t}_i))$ ($\phi(\boldsymbol{x}_i, \boldsymbol{t}_j) = \cos(\mathcal{V}(\boldsymbol{x}_i), \mathcal{T}(\boldsymbol{t}_j))$). It is noteworthy that the image and text embeddings are normalized using the $\ell_2$ norm to have a unit norm. Based on these notations, the CLIP loss can be formalized by the following:

$$\mathcal{L}_{\text{CLIP}} = -\frac{1}{2N_b} \left( \sum_{i=1}^{N_b} \log \left[ \frac{\exp(\phi(\boldsymbol{x}_i, \boldsymbol{t}_i)/\tau)}{\sum_{j=1}^{N_b} \exp(\phi(\boldsymbol{x}_i, \boldsymbol{t}_j)/\tau)} \right] \right.$$
$$\left. + \sum_{j=1}^{N_b} \log \left[ \frac{\exp(\phi(\boldsymbol{x}_j, \boldsymbol{t}_j)/\tau)}{\sum_{i=1}^{N_b} \exp(\phi(\boldsymbol{x}_i, \boldsymbol{t}_j)/\tau)} \right] \right),$$

where $\tau$ is a trainable temperature parameter.

**Zero-shot classification in CLIP.** To leverage CLIP on the downstream classification task where the input image $\boldsymbol{x} \in D_{\text{Test}}$ and class name $y_i \in \{1, 2, \cdots, c\}$, a simple yet effective way is using a class template function $T(j)$ which generates a class-specific text such as `"a photo of [CLS]"` where `[CLS]` can be replaced by the $j$-th class name on the dataset. In the inference stage, one can directly calculate the posterior probability of the image $x$ for the $i$-th class as the following:

$$p(y = i | \boldsymbol{x}) = \frac{\exp(\phi(\boldsymbol{x}, T(i))/\tau)}{\sum_{j=1}^{c} \exp(\phi(\boldsymbol{x}, T(j))/\tau)}. \qquad (1)$$

In addition, recent research (Yang et al., 2023c; Pratt et al., 2023; Maniparambil et al., 2023; Yu et al., 2023; Zhang et al., 2024; Saha et al., 2024; Feng et al., 2023b; Liu et al., 2024) delves into engineering fine-grained class-specific attributes or prompting large language models (e.g., GPT-4 (Achiam et al., 2023)) to generate attribute-related texts.

### 2.2. Backdoor Attacks and Defenses

The backdoor attack is a serious security threat to machine learning systems (Li et al., 2022; Carlini & Terzis, 2021; Xu et al., 2022; Chen et al., 2021). The whole process of a backdoor attack can be expounded as follows. In the data collection stage of a machine learning system, a malicious adversary could manufacture a part of backdoor samples with the imperceptible trigger poisoned into the training dataset. After the model training stage, the hidden trigger could be implanted into the victim model without much impact on the performance of the victim model. During the inference stage, the adversary could manipulate the victim model to produce a specific output by adding the trigger to the clean input. Early research on backdoor attacks focuses on designing a variety of triggers that satisfy the practical scenarios mainly on image and text classification tasks including invisible stealthy triggers (Chen et al., 2017; Turner

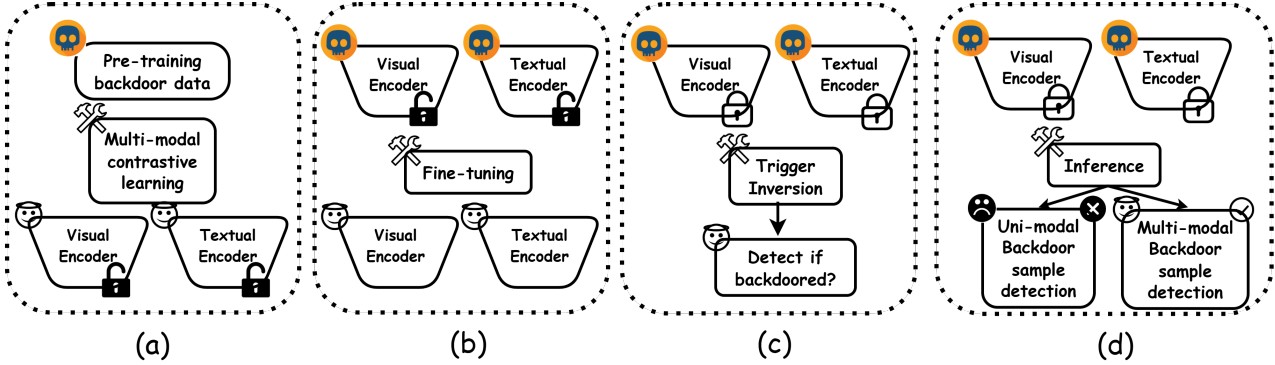

*Figure 1.* Current backdoor defense paradigms in CLIP. (a) Robust anti-backdoor contrastive learning (Yang et al., 2023b); (b) Fine-tuning a backdoored CLIP (Bansal et al., 2023); (c) Detecting a CLIP if backdoored (Sur et al., 2023; Feng et al., 2023a); (d) Our test-time backdoor sample detection. Our multimodal detection method is more effective and efficient than existing unimodal detection methods.

et al., 2019; Li et al., 2021a; Doan et al., 2021; Nguyen & Tran, 2021; Gao et al., 2023; Souri et al., 2022) and physical triggers (Chen et al., 2017; Wenger et al., 2021). To defend against these attacks, many backdoor defense methods are proposed which can be divided into four categories, mainly including data cleaning in the pre-processing stage (Tran et al., 2018), robust anti-backdoor training (Chen et al., 2022; Zhang et al., 2022), mitigation, detection, and inversion in the post-training stage (Min et al., 2023), and test-time detection in the inference stage (Shi et al., 2023).

**Backdoor attacks for CLIP.** This paper especially focuses on investigating backdoor security in multimodal contrastive learning. Recent research (Carlini & Terzis, 2021; Carlini et al., 2023; Bansal et al., 2023; Jia et al., 2022; Bai et al., 2023; Liang et al., 2023) has revealed the serious backdoor vulnerability of CLIP. Specifically, a malicious adversary can manufacture a proportion of backdoor image-text pairs $\mathcal{D}_{\mathrm{BD}} = \{(\boldsymbol{x}_i^*, T(y_t))\}_{i=1}^{N_{\mathrm{BD}}}$ where $\boldsymbol{x}_i^* = (1-\mathcal{M}) \odot \boldsymbol{x}_i + \mathcal{M} \odot \Delta$ is a backdoor image with the trigger pattern $\Delta$ (Gu et al., 2017; Chen et al., 2017) and the mask $\mathcal{M}$, and $T(y_t)$ is the caption of the target attack class $y_t$. Then, the original pre-training dataset $\mathcal{D}_{\mathrm{Train}}$ could be poisoned as $\mathcal{D}_{\mathrm{Poison}} = \{\mathcal{D}_{\mathrm{BD}} \cup \mathcal{D}_{\mathrm{Clean}}\}$. The backdoor attack for CLIP can be formalized by:

$$\{\theta_{\mathcal{V}^*}, \theta_{\mathcal{T}^*}\} = \underset{\{\theta_{\mathcal{V}}, \theta_{\mathcal{T}}\}}{\arg\min} \mathcal{L}_{\mathrm{CLIP}}(\mathcal{D}_{\mathrm{Clean}}) + \mathcal{L}_{\mathrm{CLIP}}(\mathcal{D}_{\mathrm{BD}}),$$

where $\theta_{\mathcal{V}^*}$ is the parameter of the infected visual encoder $\mathcal{V}^*(\cdot)$ and $\theta_{\mathcal{T}^*}$ is the parameter of the infected textual encoder $\mathcal{T}^*(\cdot)$. It is noteworthy that the zero-shot performance of the backdoored CLIP is expected to be unaffected in Eq. (1), while for the image $\boldsymbol{x}_i^*$ with a trigger, the posterior probability of the image for the $y_t$-th target class could be large with high probability:

$$p(y_i = y_t | \boldsymbol{x}_i^*) = \frac{\exp(\phi(\boldsymbol{x}_i^*, T(y_t))/\tau)}{\sum_{j=1}^c \exp(\phi(\boldsymbol{x}_i^*, T(j))/\tau)}. \quad (2)$$

**Defenses for the backdoored CLIP.** Effective defense methods have been proposed recently, which can be divided into three kinds of defense paradigms including anti-backdoor learning (Yang et al., 2023b) in the pre-training stage, fine-tuning the backdoored CLIP (Bansal et al., 2023; Kuang et al., 2024; Xun et al., 2024), and using trigger inversion techniques (Sur et al., 2023; Feng et al., 2023a) to detect the visual encoder of CLIP if is infected. However, due to the need for additional learning or optimization processes, these defense methods are computationally expensive. Furthermore, in many real-world scenarios, we only have access to third-party models or APIs, making it impossible to apply existing backdoor defense methods for pre-training and fine-tuning.

## 3. The Proposed Approach

In this section, we provide the first attempt at test-time backdoor detection for CLIP and propose BDetCLIP that effectively detects test-time backdoored images based on the text modality.

### 3.1. A Defense Paradigm: Test-Time Backdoor Sample Detection

Compared with existing defense methods used in the pre-training or fine-tuning stage, detecting (and then refusing) backdoor images in the inference stage directly is a more lightweight and straightforward solution to defend backdoored CLIP. To this end, one may directly adapt existing unimodal detection methods (Gao et al., 2019; Zeng et al., 2021; Udeshi et al., 2022; Guo et al., 2023; Liu et al., 2023; Pal et al., 2024; Hou et al., 2024) solely based on the visual encoder (i.e., $\mathcal{V}^*(\cdot)$) of CLIP with proper modifications. However, this strategy is *suboptimal* because of the lack of the utilization of the textual encoder $\mathcal{T}^*(\cdot)$ in CLIP to assist detection (as shown in Figure 2(a)). In contrast, we propose to integrate the visual and textual encoders in CLIP for *test-time backdoor sample detection* (TT-BSD). The objective

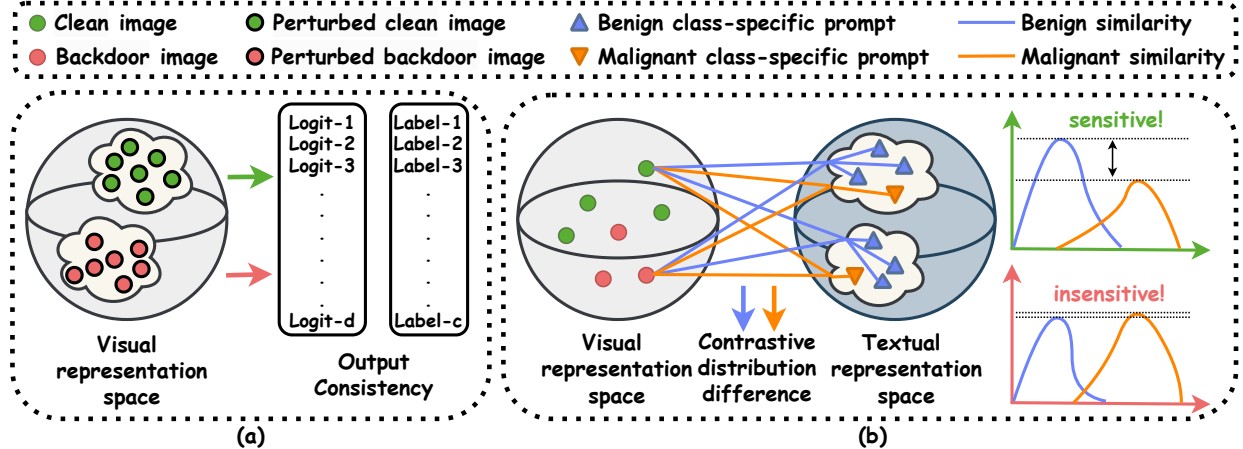

*Figure 2.* (a) Illustration of unimodal backdoor detection that only focuses on the visual representation space; (b) Illustration of BDetCLIP that leverages both image and text modalities in CLIP.

of TT-BSD for CLIP is to design a good detector $\Gamma$:

$$\Gamma = \arg\min_{\Gamma} \frac{1}{n}\Big(\sum_{\boldsymbol{x}\in\mathcal{D}_{\text{Clean}}} \mathbb{I}(\Gamma(\boldsymbol{x}, \mathcal{V}^*, \mathcal{T}^*) = 1)$$

$$+ \sum_{\boldsymbol{x}^*\in\mathcal{D}_{\text{BD}}} \mathbb{I}(\Gamma(\boldsymbol{x}^*, \mathcal{V}^*, \mathcal{T}^*) = 0)\Big), \quad (3)$$

where $\mathbb{I}(\cdot)$ is an indicator function, and $\Gamma(\boldsymbol{x})$ returns 1 or 0 indicates the detector regards $\boldsymbol{x}$ as a backdoored or clean.

**Defender's goal.** Defenders aim to design a good detector $\Gamma$ in terms of effectiveness and efficiency. Effectiveness is directly related to the performance of $\Gamma$, which can be evaluated by AUROC. Efficiency indicates the time used for detection, which is expected to be short in real-world applications.

**Defender's capability.** In this paper, we consider the *black-box* setting. Specifically, defenders can only access the encoder interface of CLIP and obtain feature embeddings of images and texts, completely lacking any prior information about the architecture of CLIP and backdoor attacks. This is a realistic and challenging setting in TT-BSD (Guo et al., 2023).

### 3.2. Our Proposed BDetCLIP

**Motivation.** It was shown that CLIP has achieved impressive zero-shot classification performance by leveraging visual description texts (Yang et al., 2023c; Pratt et al., 2023; Maniparambil et al., 2023; Yu et al., 2023; Saha et al., 2024; Feng et al., 2023b; Liu et al., 2024) generated by large language models. For backdoored CLIP (i.e., CLIP corrupted by backdoor attacks), recent research (Bansal et al., 2023) has revealed that implanted visual triggers in CLIP can exhibit a strong co-occurrence with the target class. However, such visual triggers in CLIP are usually simple non-semantic pixel patterns, which could not align well with abundant textual concepts. Therefore, backdoored images with visual

triggers are unable to properly capture the semantic changes of class description texts. This motivates us to consider whether the alignment between the visual representations of backdoored images and the class description texts would be significantly changed when there exist significant changes in the class description texts. Interestingly, we empirically find that the alignment of backdoor samples would not be significantly changed even given significant changes in the text description texts. This observation can help us distinguish backdoor samples from clean samples because the alignment of clean samples would be significantly influenced by the changes in the text description texts.

**Contrastive prompting.** Based on the above motivation, we propose BDetCLIP, a novel test-time backdoor detection method based on contrastive prompting. Specifically, we prompt GPT-4 (Achiam et al., 2023) to generate two types of contrastive class description texts. Firstly, based on the powerful in-context learning capabilities of GPT-4, we use specially designed instructions with a *demonstration* as shown in Appendix A. In particular, the demonstration for the class "goldfish" is associated with various attributes of objects, e.g., shape, color, structure, and behavior. In this way, GPT-4 is expected to output multiple fine-grained attribute-based sentences for the assigned $j$-th class, denoted by $ST_j^k(k \in [m])$ where $m$ is the number of sentences. On the other hand, we also prompt GPT-4 by the instruction "Please randomly generate a sentence of no more than 10 words unrelated to {*Class Name*}", to generate one random sentence unrelated to the assigned $j$-th class. We concatenated the class template prompt with the obtained random sentences to generate the final class-specific malignant prompt, denoted by $RT_j$, such as "A photo of a goldfish. The sun cast shadows on the bustling city street.". In Appendix D, We also recorded the money and time costs associated with the prompts generated by GPT-4, and demonstrated the

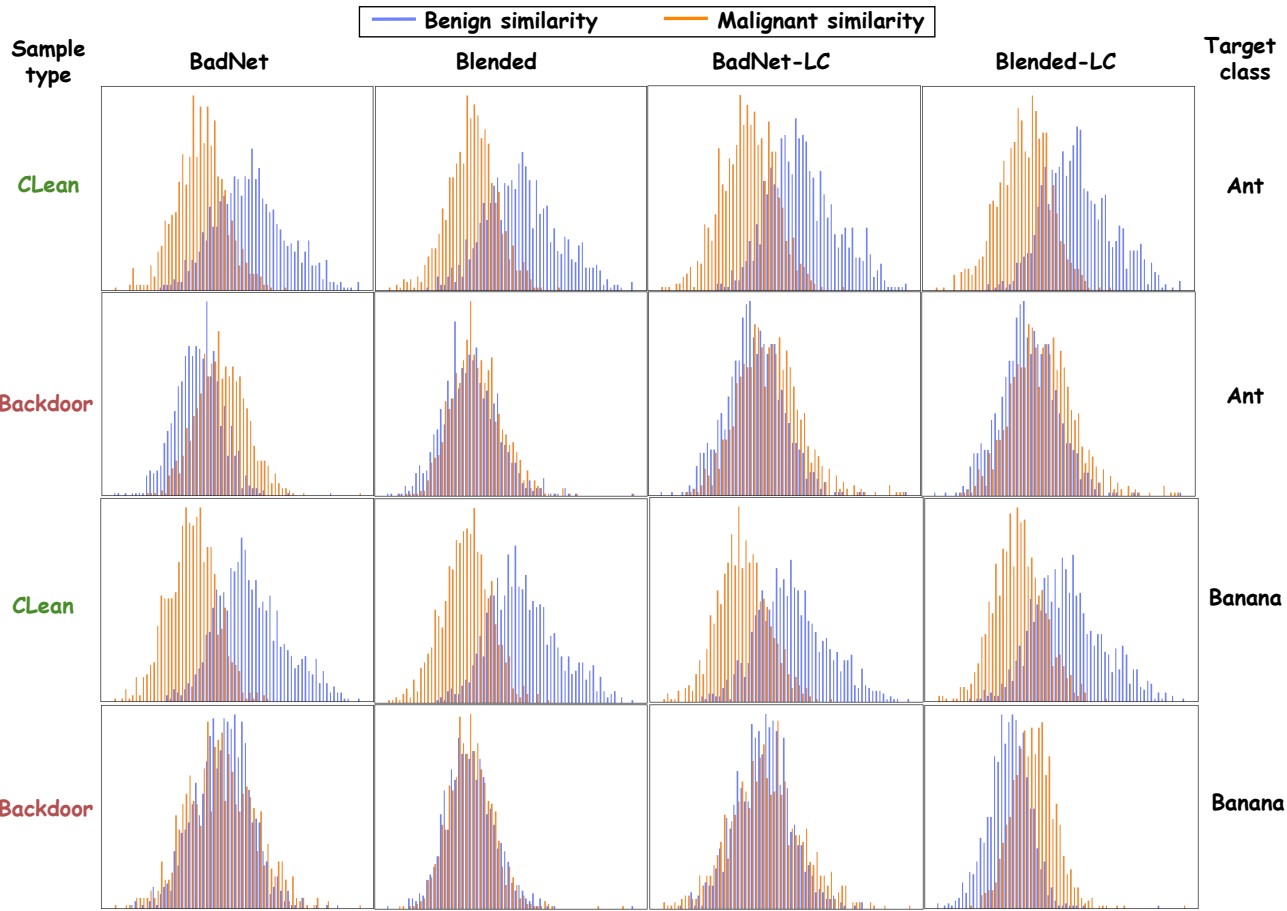

*Figure 3.* Empirical density distributions of benign and malignant similarities for 1,000 classes on ImageNet-1K. *The larger the overlap proportion in the figure, the smaller the difference in contrastive distributions.* We have omitted coordinate axes for a better view.

feasibility of using open-source models (e.g., LLaMA3-8B (Dubey et al., 2024) and Mistral-7B-Instruct-v0.2 (Jiang et al., 2023)) as alternatives.

**Contrastive distribution difference.** Based on the generated two types of texts by GPT-4, we can calculate the benign (malignant) similarity between test images and benign (malignant) class-specific prompts. In particular, we consider this calculation towards all classes in the label space, since we have no prior information about the label of each test image. In this way, we can obtain the whole distribution difference for all classes by accumulating the contrastive difference between the per-class benign and malignant similarity. Formally, for each class $y \in \mathbb{Y}$, the benign and malignant similarity for each test image $\boldsymbol{x}^t$ is denoted by $\phi(\mathcal{V}^*(\boldsymbol{x}^t), \frac{1}{m}\sum_{k=1}^{m}\mathcal{T}^*(ST_y^k))$ and $\phi(\mathcal{V}^*(\boldsymbol{x}^t), \mathcal{T}^*(RT_y))$ respectively. It is worth noting that we consider the average textual embeddings of all $m$ class-related description texts. Unless stated otherwise, the default value of parameter $m$ in our experiments is 7. Then, the contrastive distribution difference of a test image $\boldsymbol{x}$ can be

formalized by:

$$\Omega(\boldsymbol{x}) = \sum_{j \in \mathbb{Y}} \Big( \phi\big(\mathcal{V}^*(\boldsymbol{x}), \frac{1}{m}\sum_{k=1}^{m}\mathcal{T}^*(ST_y^k)\big)$$
$$- \phi\big(\mathcal{V}^*(\boldsymbol{x}), \mathcal{T}^*(RT_y)\big)\Big). \quad (4)$$

This statistic reveals the *sensitivity* of each test image towards the benign and malignant changes of class-specific prompts. We show the empirical density distributions of benign and malignant similarities on ImageNet-1K in Figure 3. In our consideration, a test-time backdoored image $\boldsymbol{x}^*$ is *insensitive* to this semantic changes of class-specific prompts, thereby leading to a relatively small value of $\Omega(\boldsymbol{x}^*)$. Therefore, we propose the following detector of TT-BSD:

$$\Gamma(\boldsymbol{x}, \mathcal{V}^*, \mathcal{T}^*) = \begin{cases} 1, & \text{if } \Omega(\boldsymbol{x}) < \epsilon, \\ 0, & \text{otherwise,} \end{cases} \quad (5)$$

where $\epsilon$ is a threshold (see Section 4.3 about how to empirically determine the value of $\epsilon$). The pseudo-code of BDet-CLIP is shown in Appendix A.

# 4. Experiments

## 4.1. Experimental setup

**Datasets.** In the experiment, we evaluate BDetCLIP on various downstream classification datasets including ImageNet-1K (Russakovsky et al., 2015), Food-101 (Bossard et al., 2014) and Caltech-101 (Fei-Fei et al., 2004). In particular, we pioneer backdoor attacks and defenses for CLIP on fine-grained image classification datasets Food-101 and Caltech-101, which are more challenging tasks. Besides, we select target backdoored samples from CC3M (Sharma et al., 2018) which is a popular multimodal pre-training dataset. During the inference stage, we set the proportion of backdoored samples in the test data to 0.3. We also provide results under different backdoor ratios (i.e., 0.3, 0.5, and 0.7) in Appendix D.8.

**Attacking CLIP.** For backdoor attacks on CLIP, we explore two approaches: pre-training CLIP from scratch on the poisoned CC3M dataset or fine-tuning a pre-trained clean CLIP using a subset of poisoned pairs. Unless otherwise specified, the models we utilize are CLIP trained on 400M samples with ResNet-50 (He et al., 2016) as the visual encoder. In terms of attack methods, our approach encompasses both traditional backdoor attack methods and advanced multimodal attack methods. On the traditional front, we utilize BadNet (Gu et al., 2017), Blended (Chen et al., 2017), LabelConsistent (Turner et al., 2019), ISSBA (Li et al., 2021b), and WaNet (Nguyen & Tran, 2021). Notably, we adapt the triggers from BadNet and Blended to execute label-consistent attacks, referred to as BadNet-LC and Blended-LC. On the multimodal attack methods, we implement BadCLIP-1 (Liang et al., 2023), BadCLIP-2 (Bai et al., 2023), TrojVQA (Walmer et al., 2022), and BadEncoder (Jia et al., 2022). Comprehensive details on the implementation of these methods are provided in Appendix B. For the target attack class, we mainly selected "banana" from ImageNet1k, "baklava" from Food-101, "dalmatian" from Caltech-101. In Appendix D.1, we provide experiments on more target attack classes. Furthermore, we explore semantic backdoor triggers and multi-target backdoor attacks in Appendix D.3 and D.4 respectively.

**Compared methods.** We cannot make a fair and direct comparison with other CLIP backdoor defense methods because our paper is the first work on backdoor detection during the inference phase for CLIP. Our method is fundamentally different from the defense methods during the fine-tuning or pre-training phases, which are designed to protect models from backdoor attacks and correct models that have been compromised by such attacks, respectively. Different from them, backdoor detection in the inference phase serves as a firewall to filter out malicious samples when we are unable to protect or correct the model. Due to the different purposes of these methods mentioned above,

their evaluation metric (i.e., ASR) is completely distinct from our evaluation metric (i.e., AUROC), making a direct comparison between our method and those methods impossible. This can be easily verified by examining the experimental settings in many recent papers focused on (unimodal) backdoor sample detection (Guo et al., 2023; Liu et al., 2023). We would like to emphasize that our BDetCLIP is applicable in the black-box setting (the defender only needs to access the output of the victim model instead of controlling the overall model), while other methods (Bansal et al., 2023; Yang et al., 2023b;a; Liang et al., 2024) have to control the whole training procedure which is infeasible in many real-world applications where only third-party models and APIs are accessible. Moreover, our defense method is much more computationally efficient, as it does not need to modify any model parameters, while previous defense methods involve the update of numerous model parameters. Given these distinctions, a direct comparison with other backdoor defense methods is not feasible. Therefore, to provide a baseline evaluation, we compare our proposed method with three widely-used unimodal test-time backdoor detection methods in conventional classification models: **STRIP** (Gao et al., 2019), **SCALE-UP** (Guo et al., 2023), and **TeCo** (Liu et al., 2023). Further implementation details can be found in Appendix B. In addition, in order to further prove the effectiveness of our method, we provide a scenario for performance comparison with other defense methods in Appendix C.

**Evaluation metrics.** Following conventional studies on backdoor sample detection, we assess defense effectiveness by using the area under the receiver operating curve (AUROC) (Fawcett, 2006). Besides, we adopt the inference time as a metric to evaluate the efficiency of the detection method. Generally, a higher value of AUROC indicates that the detection method is more *effective* and a shorter inference time indicates that the detection method is more *efficient*. We also report additional metrics such as Accuracy, Recall, and F1 in Section 4.3 to comprehensively evaluate the effectiveness of BDetCLIP.

## 4.2. Experimental results

**Overall comparison.** As shown in Table 1 and Table 2, BDetCLIP consistently outperformed the comparison methods in almost all attack settings and datasets. Specifically, BDetCLIP achieved an impressive average AUROC, exceeding 0.938 in multimodal attack scenarios and 0.972 in traditional attack scenarios, firmly establishing its superior effectiveness. In contrast, the unimodal detection methods generally exhibited significantly lower performance. Although TeCo occasionally approached BDetCLIP's performance, it demonstrated inconsistencies and performed worse overall. This highlights the ineffectiveness of unimodal methods for robust test-time backdoor detection, contrasting sharply

*Table 1.* AUROC comparison on different attacks. The best result is highlighted in bold.

| Attack→ Detection↓ | Multimodal Attack | | | | Average |
|---|---|---|---|---|---|
| | BadCLIP-1 | BadCLIP-2 | TrojVQA | BadEncoder | |
| STRIP | 0.794 | **0.987** | 0.255 | 0.341 | 0.594 |
| SCALE-UP | 0.669 | 0.976 | 0.744 | 0.694 | 0.771 |
| TeCo | 0.443 | 0.428 | 0.438 | 0.567 | 0.469 |
| BDetCLIP (Ours) | **0.900** | 0.977 | **0.978** | **0.898** | **0.938** |

| Attack→ Detection↓ | Traditional Attack | | | | | | Average |
|---|---|---|---|---|---|---|---|
| | BadNet | Blended | BadNet-LC | Blended-LC | ISSBA | WaNet | |
| STRIP | 0.772 | 0.111 | 0.803 | 0.150 | 0.351 | 0.243 | 0.405 |
| SCALE-UP | 0.737 | 0.692 | 0.690 | 0.853 | 0.515 | 0.920 | 0.735 |
| TeCo | 0.827 | 0.954 | 0.799 | 0.979 | 0.496 | 0.946 | 0.934 |
| BDetCLIP (Ours) | **0.972** | **0.983** | **0.964** | **0.997** | **0.927** | **0.989** | **0.972** |

*Table 2.* AUROC comparison on Food101 (Bossard et al., 2014) and Caltech101 (Fei-Fei et al., 2004) datasets. The best result is highlighted in bold.

| Target class | Method | BadNet | Blended | Average |
|---|---|---|---|---|
| Food101 (Baklava) | STRIP | 0.893 | 0.244 | 0.569 |
| | SCALE-UP | 0.768 | 0.671 | 0.720 |
| | TeCo | 0.834 | 0.949 | 0.892 |
| | BDetCLIP (Ours) | **0.941** | **0.977** | **0.959** |
| Caltech101 (Dalmatian) | STRIP | 0.868 | 0.672 | 0.770 |
| | SCALE-UP | 0.632 | 0.585 | 0.609 |
| | TeCo | 0.637 | 0.913 | 0.775 |
| | BDetCLIP (Ours) | **0.977** | **0.989** | **0.983** |

*Table 3.* Inference time on ImageNet-1K (Russakovsky et al., 2015). Totally 50000 test samples.

| Method | STRIP | SCALE-UP | TeCo | **BDetCLIP (Ours)** |
|---|---|---|---|---|
| Inference time | 253m 42.863s | 9m 7.066s | 637m 34.350s | **3m 8.436s** |

with the clear effectiveness advantages offered by BDetCLIP. As for efficiency, BDetCLIP also achieved the best performance for the inference time. As shown in Table 3, TeCo is the slowest detection method, even more than 160 times slower than BDetCLIP. This is because TeCo uses many time-consuming corruption operators on images which is too heavy in CLIP. This operation is also used in unimodal methods STRIP and SCALE-UP. In contrast, BDetCLIP only leverages the semantic changes in the text modality twice for backdoor detection, i.e., benign and malignant class-specific prompts. Therefore, BDetCLIP can achieve fast test-time backdoor detection in practical applications. In conclusion, BDetCLIP exhibits superior performance with respect to both effectiveness and efficiency when compared to the existing unimodal methods.

**Backdoor detection for CLIP using ViT-B/32.** We also evaluated the case where ViT-B/32 (Dosovitskiy et al., 2020) served as the visual encoder of backdoored CLIP. As shown in Table 4, our proposed BDetCLIP also achieved superior performance across all types of backdoor attacks. Con-

cretely, other methods have a significant drop in performance compared with the results in Table 1, while BDetCLIP also maintains a high level of AUROC (e.g., the average AUROC is 0.950). This observation validates the versatility of BDetCLIP in different vision model architectures of CLIP.

**Backdoor detection for backdoored CLIP pre-trained on CC3M.** Following CleanCLIP (Bansal et al., 2023), we also considered pre-training CLIP from scratch on the poisoned CC3M dataset. As shown in Table 5, compared with the results in Table 1, STRIP failed to achieve detection in almost all cases, SCALE-UP and TeCo became worse, while BDetCLIP also achieved superior performance across all attack settings, which definitely validates the versatility of BDetCLIP in different model capabilities of CLIP.

**The impact of the number of class-specific benign prompts.** As shown in Table 6, we find that increasing the number of class-specific benign prompts strengthens detection against various backdoor attacks. For example, when using 1 benign prompt, the average AUROC is 0.952,

*Table 4.* Comparison of AUROC on ImageNet-1K (Russakovsky et al., 2015), the visual encoder of CLIP is ViT-B/32 (Dosovitskiy et al., 2020). The target label of the backdoor attack is "Banana".

| Attack→
Detection↓ | BadNet | Blended | BadNet-LC | Blended-LC | Average |
|---|---|---|---|---|---|
| STRIP | 0.527 | 0.025 | 0.606 | 0.020 | 0.295 |
| SCALE-UP | 0.652 | 0.875 | 0.649 | 0.867 | 0.761 |
| TeCo | 0.714 | 0.969 | 0.727 | 0.969 | 0.845 |
| BDetCLIP (Ours) | **0.930** | **0.980** | **0.903** | **0.985** | **0.950** |

*Table 5.* Comparison of AUROC on ImageNet-1K (Russakovsky et al., 2015) dataset, the CLIP is pre-trained with CC3M (Sharma et al., 2018). The target label of the backdoor attack is "Banana".

| Attack→
Detection↓ | BadNet | Blended | Label-Consistent | Average |
|---|---|---|---|---|
| STRIP | 0.061 | 0.005 | 0.420 | 0.162 |
| SCALE-UP | 0.651 | 0.627 | 0.612 | 0.630 |
| TeCo | 0.779 | 0.782 | 0.765 | 0.775 |
| BDetCLIP (Ours) | **0.986** | **0.982** | **0.908** | **0.959** |

*Table 6.* Comparison of AUROC using different numbers of class-specific benign prompts on ImageNet-1K (Russakovsky et al., 2015). The target label of the backdoor attack is "Banana".

| Attack→
The number of class-specific benign prompts↓ | BadNet | Blended | BadNet-LC | Blended-LC | Average |
|---|---|---|---|---|---|
| using 1 class-specific benign prompt | 0.927 | 0.984 | 0.901 | 0.995 | 0.952 |
| using 3 class-specific benign prompts | 0.960 | 0.984 | 0.948 | 0.996 | 0.972 |
| using 5 class-specific benign prompts | 0.969 | 0.984 | 0.959 | 0.996 | 0.977 |
| using 7 class-specific benign prompts | **0.972** | **0.985** | **0.964** | **0.997** | **0.980** |

*Table 7.* Comparison of AUROC using different word counts in the class-perturbed random text on ImageNet-1K (Russakovsky et al., 2015). The target label of the backdoor attack is "Banana".

| Attack→
random sentence in class-specific malignant prompt ↓ | BadNet | Blended | BadNet-LC | Blended-LC | Average |
|---|---|---|---|---|---|
| no more than 10 words | **0.972** | **0.983** | **0.964** | **0.997** | **0.979** |
| no more than 20 words | 0.963 | 0.968 | 0.954 | 0.993 | 0.970 |
| no more than 30 words | 0.950 | 0.941 | 0.957 | 0.992 | 0.960 |

which increases to 0.980 when using 7 benign prompts. This is because more diverse fine-grained description texts expand the difference of contrastive distributions, which is more beneficial for BDetCLIP to distinguish backdoored and clean images. Therefore, it is of vital importance to leverage more diverse description texts in BDetCLIP.

**The impact of the text length of class-specific malignant prompts.** As shown in Table 7, the performance has a bit of a drop as the number of words in class-specific malignant prompts increases. This is because more random texts generated in class-specific malignant prompts would greatly destroy the semantics of class-specific malignant prompts, thereby increasing the contrastive distribution difference of backdoored images (close to that of clean images). This would degrade the performance of detection. Besides, the performance on BadNet and Blended attacks exhibit a relatively high sensitivity to the text length of class-specific malignant prompts.

### 4.3. Threshold Selection

Our proposed BDetCLIP can efficiently and effectively map input images to a linearly separable space. The defender needs to set a threshold $\epsilon$ to distinguish between clean images and backdoor images. In determining this threshold, we follow a widely used protocol in previous studies (Guo et al., 2023),(Liu et al., 2023): the defender can set a proper threshold based on a small set of clean validation data. Specifically, we first sampled clean samples at the designated sampling rates. Then, using formulation 4, we computed the contrastive distribution difference for all samples, ranked them from largest to smallest, and selected the 85th percentile as the threshold (notably, the specific threshold percentile can be adjusted based on real-world defense requirements). To assess the sensitivity of our approach, we chose three sampling ratios: 1%, 0.5%, and 0.1%. As shown in Table 8, 9 and 10, even when a very small sampling ratio is used, despite the increased standard deviation in the threshold, our method achieves exceptional performance across all metrics,

*Table 8.* The backdoor target label is ant. We use a backdoor ratio of 0.3 and a sampling rate of 1%.

| Backdoor | Accuracy | Recall | F1 | AUROC | Threshold |
|---|---|---|---|---|---|
| Badnet | 0.8941 ± 0.0107 | 0.9902 ± 0.0013 | 0.8488 ± 0.0127 | 0.9906 ± 0.0003 | 11.7225 ± 1.2723 |
| Blended | 0.8772 ± 0.0061 | 0.9279 ± 0.0142 | 0.8193 ± 0.0151 | 0.9425 ± 0.0003 | 12.0281 ± 1.2399 |
| Badnet-LC | 0.8938 ± 0.0074 | 0.9842 ± 0.0016 | 0.8476 ± 0.0088 | 0.9796 ± 0.0004 | 16.7526 ± 0.8944 |
| Blended-LC | 0.8837 ± 0.0068 | 0.9396 ± 0.0102 | 0.8290 ± 0.0067 | 0.9420 ± 0.0005 | 15.9315 ± 1.2748 |

*Table 9.* The backdoor target label is ant. We use a backdoor ratio of 0.3 and a sampling rate of 0.5%.

| Attack | Accuracy | Recall | F1 | AUROC | Threshold |
|---|---|---|---|---|---|
| Badnet | 0.8950 ± 0.0160 | 0.9903 ± 0.0013 | 0.8502 ± 0.0189 | 0.9908 ± 0.0003 | 11.6161 ± 1.8596 |
| Blended | 0.8772 ± 0.0109 | 0.9224 ± 0.0211 | 0.8186 ± 0.0096 | 0.9416 ± 0.0003 | 11.7094 ± 1.9545 |
| Badnet-LC | 0.8958 ± 0.0128 | 0.9835 ± 0.0029 | 0.8501 ± 0.0151 | 0.9797 ± 0.0004 | 16.4568 ± 1.5488 |
| Blended-LC | 0.8865 ± 0.0070 | 0.9347 ± 0.0106 | 0.8317 ± 0.0070 | 0.9422 ± 0.0005 | 15.3494 ± 1.3340 |

*Table 10.* The backdoor target label is ant. We use a backdoor ratio of 0.3 and a sampling rate of 0.1%.

| Attack | Accuracy | Recall | F1 | AUROC | Threshold |
|---|---|---|---|---|---|
| Badnet | 0.8775 ± 0.0107 | 0.9904 ± 0.0040 | 0.8312 ± 0.0418 | 0.9905 ± 0.0004 | 13.0927 ± 4.3069 |
| Blended | 0.8564 ± 0.0315 | 0.9391 ± 0.0430 | 0.7987 ± 0.0279 | 0.9424 ± 0.0003 | 14.2042 ± 4.4056 |
| Badnet-LC | 0.8799 ± 0.0453 | 0.9831 ± 0.0082 | 0.8341 ± 0.0488 | 0.9795 ± 0.0005 | 17.6179 ± 4.8725 |
| Blended-LC | 0.8722 ± 0.0269 | 0.9404 ± 0.0363 | 0.8167 ± 0.0524 | 0.9422 ± 0.0004 | 16.9313 ± 4.2465 |

particularly in terms of recall, due to its high AUROC value, which demonstrates its strong discriminative capability.

### 4.4. Additional experiments

In Appendix D.1, we present detection results for more backdoor target categories on ImageNet, along with results under different backdoor ratios (i.e., 0.3, 0.5, and 0.7) in Appendix D.8, both demonstrating the stability and effectiveness of our method. In Appendix D.2, D.3, and D.4, we present results for open-set detection, semantic trigger, and multi-target attack, all confirming our method's state-of-the-art performance. Appendix D.5 provides the time and money cost of GPT-4 prompt generation, highlighting the efficiency and cost-effectiveness of our method. We also test eight open-source LLMs for prompt generation in Appendix D.6, which demonstrates that open-source models can serve as stable prompt generators, replacing GPT-4 in generating both benign and malicious prompts. Furthermore, we investigate the relationship between thresholds and the number of benign prompts m in Appendix D.7.

### 5. Conclusion

In this paper, we provided the first attempt at a computationally efficient backdoor detection method to defend against backdoored CLIP in the inference stage. We empirically observed that the visual representations of backdoored images are insensitive to significant changes in class description texts. Motivated by this observation, we proposed a novel test-time backdoor detection method based on contrastive prompting, which is called BDetCLIP. For our proposed BDetCLIP, we first prompted the language model (e.g., GPT-4) to produce class-related description texts (benign) and class-perturbed random texts (malignant) by specially designed instructions. Then, we calculated the distribution difference in cosine similarity between images and the two types of class description texts, and utilized this distribution difference as the criterion to detect backdoor samples. Comprehensive experimental results validated that our proposed BDetCLIP is more effective and more efficient than state-of-the-art backdoor detection methods.

### 6. Limitations

The main limitation of this work lies in that only the CLIP model is considered because existing backdoor research on multimodal contrastive learning commonly considers CLIP as a representative victim model due to its reproducibility. In addition, our employed strategy to determine the threshold $\epsilon$ is relatively simple. More effective strategies could be further proposed to obtain a more suitable threshold.

### 7. Future work

We aim to discuss future work from both offensive and defensive perspectives. For more sophisticated backdoor attacks, we propose designing triggers that can naturally adapt to changes in prompt semantics, thereby creating more covert backdoor attacks. For enhanced backdoor defense, we suggest developing a framework for evaluating prompt quality to further improve the quality of prompts.

## Acknowledgement

Feng Liu is supported by the Australian Research Council (ARC) with grant number DE240101089, LP240100101, DP230101540 and the NSF&CSIRO Responsible AI program with grant number 2303037.

## Impact Statement

Our research contributes to AI security by detecting backdoor samples in the inference phase, which has a positive social impact. However, we acknowledge the possibility that sophisticated attackers could use our insight to bypass our defense to threaten AI security. Future work should explore the robustness of our method against adaptive attacks.

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

---

**Prompt used to generate class-related description texts:**

    I am creating class attributes for a zero-shot image recognition algorithm to classify different images. The attributes are part of the fine-grained information about the classesThis information must be deeply related to the category, and cannot be some low-quality information, such as goldfish are living things, goldfish have life, and so on.
    For example, if I say what attributes help us identify goldfish? You should respond:
        "goldfish":[
            "Goldfish are known for their bright orange or gold color but they can also come in other colors like white, black, red, and yellow.",
            "Goldfish have a variety of body shapes, ranging from the common slim-bodied type  to more rounded or egg-shaped varieties.",
            "Goldfish typically have a single dorsal fin, paired pectoral and pelvic fins, and a forked caudal (tail) fin. Some varieties, like the fancy goldfish, may have long, flowing fins."
            "Most goldfish have shiny, metallic scales, but some varieties, like the pearl scale goldfish, have uniquely textured scales."
            "Goldfish are known for their active swimming behavior and are often seen exploring their environment."]

Now I want to ask you: What attributes help us identify {Class Name}?

---

**Prompt used to generate class-perturbed random texts:**

Please randomly generate a sentence of no more than 10 words unrelated to {Class Name}

**GPT-4 OUTPUT Example (Class Name: goldfish):**

The bright sun cast shadows on the bustling city street.

---

*Figure 4.* Prompts for generating class-related description texts and class-perturbed random texts.

## A. Prompt Design

Generative Pretrained Large Language Models, such as GPT-4, have been demonstrated (Yang et al., 2023c; Pratt et al., 2023; Maniparambil et al., 2023; Yu et al., 2023; Saha et al., 2024; Feng et al., 2023b; Liu et al., 2024) to be effective in generating visual descriptions to assist CLIP in classification tasks for the following reasons: (1) These models are trained on web-scale text data, encompassing a vast amount of human knowledge, thereby obviating the need for domain-specific annotations. (2) They can easily be manipulated to produce information in any form or structure, making them relatively simple to integrate with CLIP prompts.

In our study, we harnessed the in-context learning capabilities of GPT-4 to generate two types of text—related description text and class-perturbed description text. The prompts used for generating the text are illustrated in Figure 4.

## B. More Details about the Experimental Setup

**Details of traditional attack.**    Following the attack setting in CleanCLIP (Bansal et al., 2023), we consider two types of attack means for CLIP, including fine-tuning pre-trained clean CLIP [1] on the part of backdoored image-text pairs from CC3M and pre-training backdoored CLIP by the poisoned CC3M dataset. In the first case, we randomly select 500,000 image-text pairs from CC3M as the fine-tuning dataset among which we also randomly select 1,500 of these pairs as target backdoor samples and apply the trigger to them while simultaneously replacing their corresponding captions with the class template for the target class. Then, we can fine-tune CLIP with the backdoored dataset. We finetune the pretrained model for 5 epochs with an initial learning rate of 1e-6 with cosine scheduling and 50 warmup steps and use AdamW as the optimizer. In the second case, following the attack setting in CleanCLIP (Bansal et al., 2023), we randomly select 1,500 image-text pairs from CC3M as target backdoor samples. Then, we pre-train CLIP from scratch on the backdoored CC3M dataset. We

---

[1]https://github.com/openai/CLIP

---

**Algorithm 1** BDetCLIP

---

**Require:** CLIP's infected visual encoder $\mathcal{V}^*(\cdot)$ and infected text encoder $\mathcal{T}^*(\cdot)$, threshold $\tau$, Test set $\mathcal{X}_{\text{test}}$; class-specific benign prompts $ST_j^k$, class-specific malignant prompts $RT_j$, cosine similarity $\phi()$.

1: **for** $\boldsymbol{x}^i$ in $\mathcal{X}_{\text{test}}$ **do**
2:     Compute benign similarity
    $\phi(\mathcal{V}^*(\boldsymbol{x}^i), \frac{1}{m}\sum_{k=1}^{m}\mathcal{T}^*(ST_j^k))$
3:     Compute malignant similarity $\phi(\mathcal{V}^*(\boldsymbol{x}^i), \mathcal{T}^*(RT_j))$

4:
$$\Omega(\boldsymbol{x}^i) \leftarrow \sum_{j=1}^{C}\left(\phi\Big(\mathcal{V}^*(\boldsymbol{x}^i), \frac{1}{m}\sum_{k=1}^{m}\mathcal{T}^*(ST_j^k)\Big)\right.$$
$$\left. - \phi\Big(\mathcal{V}^*(\boldsymbol{x}^i), \mathcal{T}^*(RT_j)\Big)\right)$$

5:     **if** $\Omega(\boldsymbol{x}^i) < \epsilon$ **then**
6:         Mark $\boldsymbol{x}^i$ as backdoored
7:     **else**
8:         Mark $\boldsymbol{x}^i$ as clean
9:     **end if**
10: **end for**
11: Output the detection results

---

trained for 64 epochs with a batch size of 128, an initial learning rate of 0.0005 for cosine scheduling, and 10000 warm-up steps for the AdamW optimizer. For the BadNet, Blended, BadNet-LC, and Blended-LC attacks, the architecture used is ResNet-50, with the target attack class being "Banana". For the WaNet attack, the architecture is ResNet-50, and the target attack class is "Goldfish". For the ISSBA attack, the architecture is ViT-B/32, and the target attack class is "Banana". All experiments are conducted on 8 NVIDIA 3090 GPUs.

**Details of multimodal attack.**

- For BadCLIP-1 (Liang et al., 2023), we used the officially provided weight files for detection. The attack target was "Banana," and the model architecture used was ResNet-50. The evaluation was conducted on ImageNet1K.

- For BadCLIP-2 (Bai et al., 2023), we followed the official setup for reproduction. BadCLIP-2 is a backdoor attack against prompt learning scenarios, which uses a learnable continuous prompt as a trigger. Although our approach is designed for CLIP that uses discrete prompts for classification tasks, we can make simple modifications to detect it. Specifically, we keep the benign prompt unchanged and modify the malignant prompt to a combination of learnable context and random text. In the experimental setup, we chose ViT-B/16 as the encoder attacked "Face," and detected it on Caltech101 using reversed contrast distribution difference.

- For BadEncoder, we adopted the officially provided weights. It is important to note that BadEncoder is not a backdoor attack targeting multimodal contrastive learning but rather a backdoor attack targeting self-supervised learning encoders. We followed the settings in the official paper, with the target attack class being "truck," and detected it on STL-10 (Coates et al., 2011) using reversed contrast distribution difference.

- TrojVQA is a dual-key backdoor attack targeting multimodal visual question-answering models. We used "SUDO" as the text trigger and a 16×16 patch as the visual trigger, with the target class being "banana." The detection was performed on ImageNet1K.

**Details of comparing methods.**

- **STRIP** (Gao et al., 2019) is the first black-box TTSD method that overlays various image patterns and observes the randomness of the predicted classes of the perturbed input to identify poisoned samples. The official open-sourced codes for STRIP (Gao et al., 2019) can be found at: https://github.com/garrisongys/STRIP. In our experiments, for each input image, we use 64 clean images from the test data for superimposition.

- **SCALE-UP** (Guo et al., 2023) is also a method for black-box input-level backdoor detection that assesses the maliciousness of inputs by measuring the scaled prediction consistency (SPC) of labels under amplified conditions, offering

effective defense in scenarios with limited data or no prior information about the attack. The official open-sourced codes for SCALE-UP (Guo et al., 2023) can be found at: `https://github.com/JunFengGo/SCALE-UP`.

- **TeCo** (Liu et al., 2023) modifies input images with common corruptions and assesses their robustness through hard-label outputs, ultimately determining the presence of backdoor triggers based on a deviation measurement of the results. The official open-sourced codes for TeCo (Liu et al., 2023) can be found at: `https://github.com/CGCL-codes/TeCo`. In our experiments, considering concerns about runtime, we selected "elastic_transform", "gaussian_noise", "shot_noise", "impulse_noise", "motion_blur", "snow", "frost", "fog", "brightness", "contrast", "pixelate", and "jpeg_compression" as methods for corrupting images. The maximum corruption severity was set to 6.

**Details of datasets.** ImageNet-1K (Russakovsky et al., 2015) consists of 1,000 classes and over a million images, making it a challenging dataset for large-scale image classification tasks. Food-101 (Bossard et al., 2014), which includes 101 classes of food dishes with 1,000 images per class, and Caltech101 (Fei-Fei et al., 2004), an image dataset containing 101 object categories and 1 background category with 40 to 800 images per category, are both commonly used for testing model performance on fine-grained classification and image recognition tasks. In our experiment, we utilized the validation set of ImageNet-1K (Russakovsky et al., 2015), along with the test sets of Food-101 (Bossard et al., 2014) and Caltech101 (Fei-Fei et al., 2004). By using a fixed backdoor ratio (0.3) on different downstream datasets in the evaluation, there are 15,000 (out of 50,000) backdoored images on ImageNet-1K, 7,575 (out of 25,250) backdoored images on Food-101, and 740 (out of 2,465) backdoored images on Caltech-101. Moreover, we also use larger backdoor ratios (0.5 and 0.7) on ImageNet-1K, resulting in 25,000 and 35,000 backdoor samples respectively.

## C. Defense results comparison with other defend methods

To facilitate a direct comparison of defense effectiveness, we made the necessary modifications. Specifically, during the inference stage, we set the backdoor ratio to 1. In BDetCLIP, samples with distribution differences below the threshold are directly discarded. The Attack Success Rate (ASR) is then calculated as the ratio of successfully attacked backdoor samples to the total number of backdoor samples. We argue that this strategy is reasonable in practical scenarios. To demonstrate the reliability and stability of our experimental results, we used the threshold selection method described in Section 4.3, performed random sampling ten times, and calculated both the mean and the standard deviation. For our detection experiments, we utilized the backdoored model provided by CleanCLIP (Bansal et al., 2023) as the victim model and compared the defense performance with the results reported in CleanCLIP (Bansal et al., 2023). As shown in Table 11, BDetCLIP can effectively decrease the ASR compared with the current fine-tuning defense method CleanCLIP (Bansal et al., 2023), proving that our BDetCLIP could be used to defend against backdoor attacks effectively in practical applications. To further validate the effectiveness of our method against state-of-the-art backdoor attacks targeting CLIP, we employed the compromised model weights provided by BadCLIP (Liang et al., 2023) as our defense testing subject. BadCLIP (Liang et al., 2023) represents the most advanced backdoor attack method specifically designed for CLIP. In our experiments, we compared our approach with baseline methods including CleanCLIP (Bansal et al., 2023), CleanerCLIP (Xun et al., 2024), and PAR (Singh et al., 2024). We also apply RoCLIP (Yang et al., 2023b) to the fine-tuning stage to defend against BadCLIP. Specifically, we follow the official settings of RoCLIP, using BadCLIP polluted data as fine-tuning data, and fine-tuning 24 epochs. We also tried SAFECLIP (Yang et al., 2023a) and we found that it could not produce effective defenses, and after the defense was made, the zero-shot classification capability of SAFECLIP was close to 0. As shown in Table 12, the experimental results demonstrate that our method achieves superior defensive performance compared to all baseline approaches.

*Table 11.* Comparison with the Defense Results of CleanCLIP. The metric is ASR.

| Attack | CleanCLIP | **BDetCLIP (ours)** |
|---|---|---|
| Badnet | 0.1046 | **0.0011 ± 0.0003** |
| Blended | 0.0980 | **0.0003 ± 0.0001** |
| Label Consistent | 0.1108 | **0.1085 ± 0.0156** |

## D. More Experimental Results

### D.1. Backdoor attack for more target classes

We conducted more BadNet attacks on the following classes from ImageNet: "Goldfish," "Lion," "Rooster," "Tench," "Basketball," and "Ant." The detection results are presented in 13, demonstrating that our method consistently achieved the

*Table 12.* Comparison with the Defense Results on BadCLIP(Liang et al., 2023). The metric is ASR.

| Method | ASR |
|---|---|
| CleanCLIP | 0.9196 |
| CleanerCIP | 0.2808 |
| PAR | 0.312 |
| RoCLIP | 0.1330 |
| **BDetCLIP (ours)** | **0.0444 ± 0.0110** |

best detection performance across all cases.

*Table 13.* Detection performance of different target classes

| Category | SCALE-UP | BDetCLIP (ours) |
|---|---|---|
| Goldfish | 0.781 | **0.977** |
| Lion | 0.806 | **0.992** |
| Rooster | 0.741 | **0.951** |
| Tench | 0.673 | **0.992** |
| Basketball | 0.750 | **0.984** |
| Ant | 0.690 | **0.990** |

### D.2. Backdoor detection for open-set detection.

We have conducted additional experiments to validate the effectiveness of our proposed BDetCLIP for open-set classification tasks. Specifically, we added a subset of Caltech-101 as the open set to ImageNet1K and set the backdoor ratio to 0.3. Table 14 shows that our proposed BDetCLIP can also achieve impressive performance on the open-set classification task, which verifies the transferability of our proposed BDetCLIP to other tasks in VLMs.

*Table 14.* Detection performance on the open-set classification task.

| Backdoor | AUROC |
|---|---|
| BadNet | 0.933 |
| Blended | 0.936 |
| BadNet-LC | 0.929 |
| Blended-LC | 0.991 |

### D.3. Backdoor detection for semantically meaningful trigger.

We have considered the scenario where the backdoor trigger has semantic meaning. Specifically, we used the popular "Hello Kitty" as a trigger and we also achieve good detection results in Table 15.

### D.4. Backdoor detection for multi-targets attack.

We have conducted more experiments about using BDetCLIP to defend against multi-target attacks. Specifically, to achieve the multi-target attack, we poisoned 1,000 (out of 500,000) samples for each target class (i.e., "goldfish", "basketball", and "banana") respectively. We fine-tuned the CLIP based on the poisoned dataset (the backdoor ratio is 0.3.) following the original experimental setting. Then, we used BDetCLIP to detect the backdoored CLIP. Table 16 shows that our BDetCLIP can still achieve impressive detection performance against the multi-target attack.

### D.5. Cost and Time Efficiency of Prompt Generation

We recorded the time and monetary costs associated with generating two types of prompts for each class in the Food-101 dataset using GPT-4 and GPT-4o. The results are summarized in Table 17. The results indicate that utilizing GPT-4 (or GPT-4o) for prompt generation is both efficient and cost-effective. Moreover, the prompt generation process can be conducted offline (prior to test-time detection), allowing the generated prompts to be directly employed in BDetCLIP for real-time detection tasks. Consequently, concerns regarding the runtime of the prompt generation step are minimal.

*Table 15.* The detection performance of Backdoor Attacks with semantically meaningful triggers ("Hello Kitty").

| SCALE-UP | BDetCLIP (ours) |
| --- | --- |
| 0.6111 | **0.8554** |

*Table 16.* The detection performance of Multi-target Attacks.

| SCALE-UP | BDetCLIP (ours) |
| --- | --- |
| 0.5404 | **0.9858** |

*Table 17.* Run Time and Money Cost by using GPT-4 or GPT-4o.

| GPT-4 | | |
| --- | --- | --- |
| **Category** | **Run Time** | **Money Cost** |
| Benign | 15m19s | 2.38 $ |
| Malignant | 2m5s | 0.12 $ |
| **GPT-4o** | | |
| **Category** | **Run Time** | **Money Cost** |
| Benign | 5m33s | 0.42 $ |
| Malignant | 1m24s | 0.06 $ |

## D.6. Using open-source models for prompts generation

We also explored the feasibility of replacing GPT-4 for prompt generation with open-source models. We utilized the following models: Llama3-8B-Instruct (Dubey et al., 2024), Mistral-7B-Instruct-v0.2 (Jiang et al., 2023), Yi-1.5-9B-Chat (Young et al., 2024), gemma-2-2b-it (Team et al., 2024), gemma-2-9b-it (Team et al., 2024), Phi-3.5-mini-instruct (Abdin et al., 2024), Qwen2.5-7B-Instruct (Yang et al., 2024), and Qwen2.5-14B-Instruct (Yang et al., 2024). The experimental results of using these varying open-source language models were recorded in Table 18. Additionally, we documented the prompt generation times for Llama3-8B-Instruct (denoted as "L") and Mistral-7B-Instruct-v0.2 (denoted as "M") in Table 19.

Although using open-source models for prompt generation may require more time (which minimally impacts detection efficiency when performed offline), the detection performance remains comparable to that achieved with GPT-4. This indicates that using open-source models is a promising alternative for prompt generation.

## D.7. The relationship between the number of benign prompts and threshold.

We denote the number of class-specific benign prompts as $m$. We conducted tests with $m = 6, 5, 4, 3$, applying the aforementioned threshold selection strategy detailed in Section 4.3. Random sampling was performed ten times for each case. Subsequently, we calculated both the variance and the mean of the selected thresholds. The mean value was then employed as the threshold for subsequent experiments. As shown in 21, We can see that the larger m is, the better the overall effect will be, and the threshold will be correspondingly larger. This is intuitive: as m increases, the number of benign prompts grows, providing more fine-grained information, which increases the semantic differences between benign prompts and malicious prompts.

## D.8. Varying ratios of test-time backdoor samples.

We conducted a comparative analysis between SCALE-UP and our method to explore the effects of variations in backdoor proportions on our efficacy. Results can be found in Table 22, 23, and 24. The results indicate that under different proportions of test-time backdoor samples, our method (BDetCLIP) consistently outperforms the baseline method SCALE-UP. Whether at a backdoor sample ratio of 0.3, 0.5, or 0.7, BDetCLIP achieves higher AUROC scores across all target categories and attack detection scenarios compared to SCALE-UP. This suggests that BDetCLIP exhibits higher robustness and accuracy in detecting backdoor samples, thereby enhancing the reliability and security of multi-modal models against backdoor attacks.

*Table 18.* Detection performance by using different models

| Model | BadNet | Blended |
|---|---|---|
| GPT-4 | 0.972 | 0.983 |
| Llama3-8B-Instruct | 0.947 | 0.983 |
| Mistral-7B-Instruct-v0.2 | 0.983 | 0.963 |
| Yi-1.5-9B-Chat | 0.977 | 0.960 |
| gemma-2-2b-it | 0.979 | 0.954 |
| gemma-2-9b-it | 0.970 | 0.948 |
| Phi-3.5-mini-instruct | 0.917 | 0.947 |
| Qwen2.5-7B-Instruct | 0.929 | 0.948 |
| Qwen2.5-14B-Instruct | 0.923 | 0.969 |
| STRIP | 0.893 | 0.244 |
| SCALE-UP | 0.768 | 0.671 |
| TeCo | 0.834 | 0.949 |

*Table 19.* Time spent on generating prompts

| Model | Benign | Malignant |
|---|---|---|
| L | 24m20s | 4m6s |
| M | 21m14s | 4m16s |

*Table 21.* Performance for Different Values of $m$.

| $m$ | Threshold (mean) | Accuracy | Recall | F1 | AUROC |
|---|---|---|---|---|---|
| 6 | 11.7199 | 0.8785 | 0.9238 | 0.8200 | 0.9417 |
| 5 | 5.2971 | 0.8640 | 0.8638 | 0.7919 | 0.9280 |
| 4 | 2.1915 | 0.8539 | 0.8335 | 0.7737 | 0.9200 |
| 3 | -1.3766 | 0.8424 | 0.7986 | 0.7523 | 0.9099 |

## D.9. Zero-shot performance and attack success rate (ASR) of using different prompts for the attacked models.

We also examined the zero-shot classification performance of CLIP subjected to a backdoor attack using our class-specific benign prompt, class-specific malignant prompt, and the original class template prompt for benign images, as well as the severity of its susceptibility to malicious images. Detailed results are presented in Table 25 and 26. The results show that when using class template prompts, the model's zero-shot performance is higher, but the attack success rate is also the highest, indicating that while these prompts offer the best classification performance, they are the most susceptible to triggering backdoor attacks. class-specific benign prompts exhibit some variability in reducing the attack success rate, with slightly lower zero-shot performance. class-specific malignant prompts generally significantly reduce the attack success rate, though their zero-shot performance is the lowest, indicating that these prompts have potential to reduce the attack success rate but at the cost of some classification performance. Overall, the choice of prompts plays a significant role in mitigating backdoor attacks, and further research in prompt engineering to enhance model robustness while maintaining high performance is a promising direction.

*Table 22.* AUROC comparison on ImageNet-1K (Russakovsky et al., 2015). The proportion of test-time backdoor samples is 0.3. The best result is highlighted in bold.

| Target class | Attack→ Detection↓ | BadNet | Blended | BadNet-LC | Blended-LC | Average |
|---|---|---|---|---|---|---|
| Ant | SCALE-UP | 0.740 | 0.670 | 0.715 | 0.737 | 0.716 |
| | BDetCLIP (Ours) | **0.993** | **0.972** | **0.984** | **0.963** | **0.978** |
| Banana | SCALE-UP | 0.737 | 0.692 | 0.690 | 0.853 | 0.743 |
| | BDetCLIP (Ours) | **0.972** | **0.983** | **0.964** | **0.997** | **0.979** |
| Basketball | SCALE-UP | 0.741 | 0.715 | 0.755 | 0.650 | 0.715 |
| | BDetCLIP (Ours) | **0.991** | **0.972** | **0.994** | **0.995** | **0.988** |

*Table 23.* AUROC comparison on ImageNet-1K (Russakovsky et al., 2015). The proportion of test-time backdoor samples is 0.5. The best result is highlighted in bold.

| Target class | Attack→ Detection↓ | BadNet | Blended | BadNet-LC | Blended-LC | Average |
|---|---|---|---|---|---|---|
| Ant | SCALE-UP | 0.737 | 0.668 | 0.714 | 0.734 | 0.713 |
| | BDetCLIP (Ours) | **0.993** | **0.972** | **0.984** | **0.963** | **0.978** |
| Banana | SCALE-UP | 0.738 | 0.693 | 0.688 | 0.854 | 0.743 |
| | BDetCLIP (Ours) | **0.972** | **0.983** | **0.964** | **0.997** | **0.979** |
| Basketball | SCALE-UP | 0.740 | 0.714 | 0.755 | 0.650 | 0.715 |
| | BDetCLIP (Ours) | **0.991** | **0.972** | **0.994** | **0.995** | **0.988** |

*Table 24.* AUROC comparison on ImageNet-1K (Russakovsky et al., 2015). The proportion of test-time backdoor samples is 0.7. The best result is highlighted in bold.

| Target class | Attack→ Detection↓ | BadNet | Blended | BadNet-LC | Blended-LC | Average |
|---|---|---|---|---|---|---|
| Ant | SCALE-UP | 0.738 | 0.670 | 0.711 | 0.735 | 0.714 |
| | BDetCLIP (Ours) | **0.992** | **0.972** | **0.984** | **0.962** | **0.978** |
| Banana | SCALE-UP | 0.738 | 0.692 | 0.689 | 0.852 | 0.743 |
| | BDetCLIP (Ours) | **0.971** | **0.984** | **0.964** | **0.997** | **0.979** |
| Basketball | SCALE-UP | 0.741 | 0.714 | 0.756 | 0.652 | 0.716 |
| | BDetCLIP (Ours) | **0.991** | **0.972** | **0.994** | **0.995** | **0.988** |

*Table 25.* Zero-shot performance of using different prompts for the attacked models.

| Target class | Attack→ Prompts↓ | BadNet | Blended | BadNet-LC | Blended-LC |
|---|---|---|---|---|---|
| Ant | class template | 0.539 | 0.540 | 0.539 | 0.537 |
| | class-specific benign prompt | 0.483 | 0.475 | 0.478 | 0.472 |
| | class-specific malignant prompt | 0.290 | 0.309 | 0.309 | 0.298 |
| Banana | class template | 0.539 | 0.537 | 0.541 | 0.538 |
| | class-specific benign prompt | 0.481 | 0.477 | 0.478 | 0.475 |
| | class-specific malignant prompt | 0.269 | 0.272 | 0.280 | 0.273 |
| Basketball | class template | 0.535 | 0.542 | 0.542 | 0.538 |
| | class-specific benign prompt | 0.474 | 0.474 | 0.477 | 0.477 |
| | class-specific malignant prompt | 0.285 | 0.278 | 0.288 | 0.298 |

*Table 26.* Attack success rate (ASR) of using different prompts for the attacked models.

| Target class | Attack→ Prompts↓ | BadNet | Blended | BadNet-LC | Blended-LC |
|---|---|---|---|---|---|
| Ant | class template | 0.983 | 0.993 | 0.971 | 0.994 |
| | class-specific benign prompt | 0.821 | 0.885 | 0.752 | 0.905 |
| | class-specific malignant prompt | 0.840 | 0.847 | 0.116 | 0.309 |
| Banana | class template | 0.985 | 0.998 | 0.974 | 0.994 |
| | class-specific benign prompt | 0.021 | 0.932 | 0.004 | 0.862 |
| | class-specific malignant prompt | 0.821 | 0.781 | 0.785 | 0.601 |
| Basketball | class template | 0.990 | 0.980 | 0.987 | 0.997 |
| | class-specific benign prompt | 0.962 | 0.856 | 0.808 | 0.917 |
| | class-specific malignant prompt | 0.716 | 0.689 | 0.806 | 0.948 |

