# OpenReview forum: "Test-Time Multimodal Backdoor Detection by Contrastive Prompting"
_ICML.cc/2025/Conference — ICML 2025 poster_

### Official Review · Reviewer_NuQR · 2025-02-27

**Overall Recommendation:** 3

**Summary:**

In this paper, the authors propose an inference-time, multi-modal backdoor detection method called BDetCLIP. The method is motivated by the observation that the visual representations of backdoored images exhibit limited sensitivity to significant changes in class description texts. Specifically, BDetCLIP first prompts GPT-4 to generate class-related description texts (benign) and class-perturbed random texts (malignant) using specially designed instructions. The method then calculates the distribution difference in cosine similarity between the images and the two types of class description texts, using this difference as the criterion to detect backdoor samples. The experimental results demonstrate that BDetCLIP is both more effective and efficient compared to other detection methods.

**Claims And Evidence:**

Yes

**Essential References Not Discussed:**

No

**Experimental Designs Or Analyses:**

There are some issues:
1. The backdoor ratios used in the experiments (i.e., 0.3, 0.5, 0.7, as listed in Appendix E.8) are excessively large. Given that training or fine-tuning a multi-modal model requires a substantial number of training samples, these large ratios imply that millions of training samples could be poisoned, which is both unrealistic and impractical.
2. The traditional backdoor attacks employed seem outdated. It would be beneficial for the authors to incorporate more recent and sophisticated attacks.

**Methods And Evaluation Criteria:**

There are some problems:
1. BDetCLIP relies on a pre-selected threshold to distinguish between clean and backdoor images, which could impact its practicality. On one hand, the defender may not be able to predict the type of backdoor attack in advance or may face adaptive attackers in real-world scenarios. In such cases, can the defender still accurately select the threshold? On the other hand, the size of the auxiliary dataset used to select the threshold seems excessively large (e.g., 1%, 0.5%, and 0.1%, as evaluated in Appendix B) for practical black-box detection tasks. Would the defender still be able to choose an appropriate threshold with a smaller auxiliary dataset, or even a few auxiliary samples?
2. The authors predominantly present results using averaged metrics such as AUROC and accuracy. However, for detection tasks, it is crucial to focus on TPR@LowFPR (e.g., TPR@0.1%FPR), as this better reflects the defender's confidence in distinguishing clean and backdoor images.

**Other Comments Or Suggestions:**

None

**Other Strengths And Weaknesses:**

None

**Questions For Authors:**

1. Can the defender still effectively select the threshold when faced with unknown or adaptive attacks?
2. Could the defender still accurately select the threshold with a smaller auxiliary dataset, or even just a few auxiliary samples?
3. The backdoor ratios in the experiments are too large and unrealistic. How will the proposed method perform when the backdoor ratio is small?

**Relation To Broader Scientific Literature:**

BDetCLIP enables inference-time backdoor detection on CLIP, eliminating the need for additional learning or optimization procedures, unlike previous detection methods.

**Theoretical Claims:**

There are no theoretical claims

---

> ### Author Rebuttal · Authors · 2025-04-01
>
> Thanks for your valuable comments. We provide point-by-point responses to your concerns as follows. We are also willing to discuss with you in the discussion period if there is anything unclear.
>
> **Q1: Unpredictable attacks and the need for large auxiliary datasets.**
>
> **A:** We would like to clarify that our method does not require prior knowledge of the backdoor attack type. The threshold is estimated using only clean images. Furthermore, the auxiliary dataset consists of 50,000 images; therefore, 0.1\% corresponds to 50 images, representing a relatively small sample size. To further address your concern, we conducted experiments using a 0.05\% sampling ratio:
> **Table: 0.05 sampling rate**
> | Metric | Mean   | Std. Dev. |
> |:--:|:--:|:--:|
> | Accuracy   | 0.8883 | 0.0492    |
> | Recall     | 0.9808 | 0.0131    |
> | F1 Score   | 0.8445 | 0.0530    |
> | AUROC      | 0.9754 | 0.0003    |
>
> **Q2: More metrics.**
>
> **A:** Thank you for your suggestion. To better show the metrics, we have provided the AUROC curves in the anonymous link: https://drive.google.com/drive/folders/109V8xGsbnVcUgRXzh6r3qM7ae3qOCrwT?usp=sharing, which includes TPR and FPR. From these images, we can see that when FPR is low, the performance still maintains a high level, which validates the method's confidence in distinguishing clean and backdoor samples.
>
> **Q3: The backdoor ratios used in the experiments.**
>
> **A:** We would like to clarify that the backdoor ratio mentioned in Appendix E.8 refers to the proportion of poisoned samples in the testing set during detection, not the poisoning ratio in the training set. We can effectively poison the model with only 1500 poisoned samples. Please refer to Appendix C, 'More Details about the Experimental Setup,' specifically the 'Details of traditional attack' section.
>
> **Q4: The traditional backdoor attacks employed seem outdated. It would be beneficial for the authors to incorporate more recent and sophisticated attacks.**
>
> **A:** We would like to explain that we have compared many advanced backdoor attacks, such as BadCLIP-1, BadCLIP-2, TrojVQA, and BadEncoder. We believe that the experimental results have sufficiently validated the superiority of our method against these recent advanced attacks.
>
> **Q5: Can the defender still effectively select the threshold when faced with unknown or adaptive attacks?**
>
> **A:**  Thank you for your review! As stated in response to Q1, our method does not require prior knowledge of the attack type. Furthermore, BadCLIP is indeed an adaptive attack, where the visual trigger closely aligns with class-specific texts, including target-class-related attribute words in the embedding space. The defense performance against BadCLIP, as shown in Table 1, demonstrates the effectiveness of our method even against adaptive attacks.
>
> **Q6: Could the defender still accurately select the threshold with a smaller auxiliary dataset or even just a few auxiliary samples?**
>
> **A:** To solve your concern, we have conducted additional experiments by only using 25 samples (sampling rate 0.05\%). The experimental results are shown in the following table.
>
> **Table: 0.05 sampling rate**
> | Metric| Mean| Std. Dev. |
> |:--:|:--:|:--:|
> | Accuracy | 0.8883 | 0.0492    |
> | Recall | 0.9808 | 0.0131    |
> | F1 Score| 0.8445 | 0.0530    |
> | AUROC | 0.9754 | 0.0003    |
>
> From the table, we can see that our method still achieves high performance, which validates its applicability in real-world applications.
>
> **Q7: The backdoor ratios in the experiments are too large and unrealistic. How will the proposed method perform when the backdoor ratio is small?**
>
> **A:** Thanks for your valuable suggestion. We have conducted additional experiments by setting the backdoor ratio to 0.01. The experimental results are shown in the following table.
>
> **Table: 0.01 backdoor ratio**
> |Attack|AUROC|
> |:--:|:--:|
> |BadNet|0.9776|
> |Blended|0.9870|
> |BadNet-LC|0.9695|
> |Blended-LC|0.9946|
>
> From the table, we can see that our method still achieves high performance even when facing the small backdoor ratio.

---

> > ### Comment · Reviewer_NuQR · 2025-04-02
> >
> > Thanks for the rebuttal. It covers my main concerns.

---

> > > ### Author Response · Authors · 2025-04-02
> > >
> > > Thank you for increasing the score, and we are delighted to know that your concerns were addressed by our rebuttal! We sincerely value your constructive feedback, and your recognition of our efforts in the paper is greatly appreciated. We note that there might be an access issue regarding our provided anonymous link, hence we updated our anonymous link to https://anonymous.4open.science/r/ICML_Rebuttal_10121/, where we provided four figures about ROC curves. From these figures, we can easily find that even when FPR is very low (e.g., 0.1), TPR is still very high. This demonstrates the effectiveness of our detection results with other evaluation metrics. Thank you again for your wonderful suggestions, which will definitely improve the quality of our paper.

---

### Official Review · Reviewer_KPXq · 2025-03-12

**Overall Recommendation:** 4

**Summary:**

BDetCLIP is a test-time multimodal backdoor detection strategy that identifies backdoor examples using a set of contrastive prompts. Specifically, the visual representation of backdoor images is less correlated to both the *benign* and *random* text description compared to clean examples, leading to a distribution difference that facilitates the defender to detect backdoor examples. The authors have conducted a series of experiments to verify the effectiveness of BDetCLIP in detecting backdoor examples and are more efficient than baseline unimodal methods.

**Claims And Evidence:**

Yes.

**Essential References Not Discussed:**

No.

**Experimental Designs Or Analyses:**

I've checked all the experiments.

**Methods And Evaluation Criteria:**

Yes, this paper applies the detection method against mainstream backdoor attacks and shows its effectiveness against other baselines.

**Other Comments Or Suggestions:**

Please see the **Questions** part.

**Other Strengths And Weaknesses:**

1. This paper is well-organized and easy to follow.
2. The proposed method is well-motivated and is effective against other baselines.
For weakness, please see the **Questions** part.

**Questions For Authors:**

1. For safety-related papers, it is encouraged to add some discussions on adaptive attacks. For instance, when conducting backdoor attacks, what if the attacker adds regularization to control the cosine similarity between poisoned images and benign text descriptions?
2. It seems that the current method is only capable of detecting the backdoors within images. What if *only* the text encoder is poisoned?
3. Is it possible to extend your method to other tasks, such as image-captioning?

**Relation To Broader Scientific Literature:**

The proposed BDetCLIP is an efficient strategy to detect backdoor examples in multimodal models, which could safeguard the usage of multimodal models.

**Theoretical Claims:**

No theoretical claims in this paper.

---

> ### Author Rebuttal · Authors · 2025-04-01
>
> Thanks for your insightful comments. We provide point-by-point responses to your concerns as follows. We are also willing to discuss with you in the discussion period if there is anything unclear.
>
> **Q1: For safety-related papers, it is encouraged to add some discussions on adaptive attacks. For instance, when conducting backdoor attacks, what if the attacker adds regularization to control the cosine similarity between poisoned images and benign text descriptions?**
>
> **A:** Thanks for your insightful suggestion. It is worth noting that BadCLIP used in the experiment is exactly one of the adaptive attacks since the triggers are optimized by attribute texts (like benign class descriptive texts) of the target class. From the experimental results shown in Table 1, we can see that our method can also achieve a high AUROC against the adaptive attack. This may be because, although the trigger is more aligned with the attributes of the target class, it is still infeasible to perceive the semantic change between benign and malignant class descriptive texts due to their diversity. We will provide more discussion on the adaptive attack and potential defense counterparts in the revised paper.
>
> **Q2: It seems that the current method is only capable of detecting the backdoors within images. What if only the text encoder is poisoned?**
>
> **A:** Thanks for throwing the interesting idea. We would like to explain that existing related research [1-7] on the backdoor security of vision-language models all focus on detecting the backdoors in the images, which is a widely used protocol, and thus we also following this setting in the paper. In addition, it is also interesting to explore your idea of detecting the textual backdoor when only the text encoder is poisoned. We will explore this idea in the future.
>
> **Q3: Is it possible to extend your method to other tasks, such as image-captioning?**
>
> **A:** Thanks for throwing the interesting idea. We believe that our BDetCLIP can be adapted to other non-classification tasks, e.g., cross-modal retrieval. For example, assume that a vision-language model has been implanted with a visual backdoor (the target class is banana), in cross-modal retrieval, a clean query image of a dog corresponds to many gallery texts related to the dog, while a query image of a dog associated with the backdoor could correspond to many gallery texts related to the banana. To defend against the backdoor attack in cross-modal retrieval, we can adapt BDetCLIP with specific modifications. Particularly, we can add benign and malignant descriptions to gallery texts, and calculate the matching differences of top-k retrieval texts. In this way, a clean query image could correspond to a changeable gallery of texts, while a backdoored query image could correspond to a stable gallery of texts. Hence, we can calculate a similar metric value indicating the stability of gallery texts and employ a similar detecting criterion in BDetCLIP. We will explore the idea in the future.
>
> **References**
>
> [1] Cleanclip: Mitigating data poisoning attacks in multimodal contrastive learning. In ICCV, 2023.
>
> [2] Badclip: dual-embedding guided backdoor attack on multimodal contrastive learning. In CVPR, 2024.
>
> [3] BadCLIP: Trigger-Aware Prompt Learning for Backdoor Attacks on CLIP. In CVPR, 2024.
>
> [4] Robust contrastive language-image pretraining against data poisoning and backdoor attacks. In NeurIPS, 2023.
>
> [5] Better Safe than Sorry: Pre-training CLIP against Targeted Data Poisoning and Backdoor Attacks. In ICML, 2024.
>
> [6] Unlearning Backdoor Threats: Enhancing Backdoor Defense in Multimodal Contrastive Learning via Local Token Unlearning. In Arxiv, 2024.
>
> [7] BadEncoder: Backdoor Attacks to Pre-trained Encoders in Self-Supervised Learning. In IEEE Symposium on Security and Privacy, 2022

---

> > ### Comment · Reviewer_KPXq · 2025-04-03
> >
> > Thanks for the response, I've no further questions.

---

> > > ### Author Response · Authors · 2025-04-06
> > >
> > > Thank you for acknowledging that our rebuttal has addressed your concerns. We are so grateful that you have kindly increased your score from 3 to 4. We sincerely thank you for your valuable comments on our paper!

---

### Official Review · Reviewer_nLrT · 2025-03-14

**Overall Recommendation:** 2

**Summary:**

The paper proposes BDetCLIP, a novel test-time backdoor detection method for CLIP models. The key insight is that backdoored images exhibit insensitivity to semantic perturbations in class description texts. The method leverages contrastive prompting:
1.	Text Generation: GPT-4 generates class-related (benign) and class-perturbed (malignant) text prompts .
2.	Similarity Analysis: Computes the distribution difference in cosine similarity between image embeddings and the two types of prompts. Backdoored images show smaller distribution shifts compared to clean images.
3.	Efficiency: Achieves faster inference than unimodal baselines (e.g., STRIP, SCALE-UP) with an average AUROC of 0.972 on traditional attacks (BadNet, Blended) across datasets (ImageNet-1K, Food-101, Caltech-101) .
The work claims contributions in (1) a new inference-stage defense paradigm for CLIP, (2) the first multimodal backdoor detection method using text-image contrast, and (3) superior empirical performance

**Claims And Evidence:**

In line 208-211, the author claimed that “However, such visual triggers in CLIP are usually simple non-semantic pixel patterns, which could not align well with abundant textual concepts.” In that case, this motivates the authors to use the alignment between the visual representations of backdoored images and the class description to design a detection method. However, for label consistent attack, the backdoored images are patched with trigger, while the captions are left unchanged. The image semantic and text semantic are still aligned. This type of attack violates the assumption of motivation in this paper, I am not sure how this work can eventually counter this attack.
The model uses a large language model GPT-4 during the detection phase, which raises an issue. If an attacker conducts adaptive attack training specifically targeting this large language model advance, they might evade this method's detection. This issue requires further discussion and exploration.
Unfair Comparison: The compared methods (e.g., STRIP, TeCo) are designed for unimodal settings and lack adaptation to multimodal CLIP. Their suboptimal performance does not conclusively prove BDetCLIP’s superiority, as multimodal-specific baselines are missing.
The attack success rate (ASR) of poisoned CLIP is not reported, which raises another problem: What if the model is not fully poisoned, such as the ASR is 70%, can this method still work? It seems like this method relies on the assumption that the CLIP is fully poisoned.

**Essential References Not Discussed:**

Below are papers not discussed in the paper:
Efficient Backdoor Defense in Multimodal Contrastive Learning: A Token-Level Unlearning Method for Mitigating Threats. https://arxiv.org/abs/2409.19526.
Unlearning Backdoor Threats: Enhancing Backdoor Defense in Multimodal Contrastive Learning via Local Token Unlearning. https://arxiv.org/abs/2403.16257
TA-Cleaner: A Fine-grained Text Alignment Backdoor Defense Strategy for Multimodal Contrastive Learning. https://arxiv.org/abs/2409.17601v1

**Experimental Designs Or Analyses:**

Threshold Dependency: The threshold ϵ is set using clean validation data (e.g., 1% of samples). This assumes access to clean data, which is unrealistic in true black-box scenarios. Sensitivity analysis (Table 8-10) shows high variance in thresholds at low sampling rates, risking unreliable deployment.
Attack Diversity: In Table 13. Why “Rooster” class will reduce the BDetCLIP detection performance to 0.951, does that mean this method is not robust to different target classes and the adversary can optimize the choice of target class to evade the detection?

**Methods And Evaluation Criteria:**

Dataset Limitations: Experiments focus on poisoned CC3M dataset but omit real-world scenarios (e.g., web-scale image-caption dataset, like MSCOCO, Flickr30K, YFCC)
CLIP variants: Experiments focus on CILP model, while the effectiveness of this method with other CLIP variants is not clear, such as DeCLIP (Supervision Exists Everywhere: A Data Efficient Contrastive Language-Image Pre-training Paradigm, https://openreview.net/forum?id=zq1iJkNk3uN), ALIGN (Scaling Up Visual and Vision-Language Representation Learning With Noisy Text Supervision, https://proceedings.mlr.press/v139/jia21b.html).
Defense comparison: results with RoCLIP and SAFECLIP are missing.
(Robust Contrastive Language-Image Pre-training against Data Poisoning and Backdoor Attacks, https://openreview.net/pdf?id=ONwL9ucoYG)
(Better Safe than Sorry: Pre-training CLIP against Targeted Data Poisoning and Backdoor Attacks, https://arxiv.org/abs/2310.05862)

**Other Comments Or Suggestions:**

Figure 3: Low resolution hinders interpretation of distribution shifts.
Typos:
In line 104. Left column, the expression should be “\{(x_i,t_i)\}_{i=1}^{N_b}” since it is a batch of pairs.
In line 67. Right column. “the input image x \belong DTest”, the dataset should be “\mathcal{D}”.
In line 136. Left column. There are two left brackets but one right bracket.
In line 138. Left column. The operation “\odot” is not defined.

**Other Strengths And Weaknesses:**

Originality: This paper seems to share the same intuition as TeCO, considering the way it uses GPT-4 to expose the poisoned samples. Similarly, TeCO applies different image corruptions to induce the different behavior for poisoned sample and benign sample.
Clarity: Well-structured, but Figure 1 could better illustrate the contrastive prompting workflow.

**Questions For Authors:**

Q1: The paper claims to be the first method to detect CLIP backdoors at the inference stage, but can existing single-modal detection methods (such as STRIP, TeCo) achieve similar results through simple extensions (for example, combining text modality)? Did the author fully demonstrate the necessity of multimodal comparison?
Q2: The core assumption of the method is that "backdoor samples are less sensitive to text changes than clean samples", but is this observation universal? Can you provide a formal analysis (e.g., mutual information between visual/textual embeddings) to explain why backdoor triggers disrupt cross-modal alignment? Like I said, the label consistent backdoor attack violates the assumption of this work.
Q3: In line 636, the authors predefined fixed 85% quantile. In actual deployment, how to dynamically adjust the threshold to adapt to different data sets or attack ratios? Has the sensitivity of threshold selection to detection performance been verified?
Q4: More results are required as mentioned in methods and evaluation criteria

**Relation To Broader Scientific Literature:**

Novelty Gap: While positioned as the first test-time defense for CLIP. Recent works like CleanCLIP (CleanCLIP: Mitigating Data Poisoning Attacks in Multimodal Contrastive Learning, https://openaccess.thecvf.com/content/ICCV2023/papers/Bansal_CleanCLIP_Mitigating_Data_Poisoning_Attacks_in_Multimodal_Contrastive_Learning_ICCV_2023_paper.pdf) address similar goals (backdoor defense) but in different stages (pre-training/fine-tuning vs. inference).

**Theoretical Claims:**

The paper makes no formal theoretical claims. Hypotheses (e.g., text-insensitivity of backdoors) are empirically validated but lack theoretical grounding (e.g., why semantic alignment fails for backdoored samples).

---

> ### Author Rebuttal · Authors · 2025-04-01
>
> Thank you for your valuable comments, and we respond to your concerns as follows. We are willing to discuss with you in the discussion period.
>
> **Q1: The label consistent attack violates the assumption of the motivation.**
>
> **A:** Thanks for raising the concern. We would like to explain that label-consistent attacks do not violate the assumption of our motivation. As for label-consistent attacks, the visual trigger (in the images of ''banana'') could only align with the target class text (e.g., ''banana'') in the unchanged caption, but could not align well with abundant textual concepts that are specific property texts of the target class (e.g., yellow color, meniscus shape, and fruit toward ''banana''). Empirically, BDetCLIP achieves high AUROCs against BadNet-LC (0.983) and Blended-LC (0.997), as shown in Table 1, which also validates our motivation.
>
> **Q2: Adaptive attacks.**
>
> **A:** Actually, BadCLIP-1 is exactly one of the adaptive attacks since the trigger used in BadCLIP-1 is optimized by using attribute texts of target classes like benign class descriptive texts used in our method.
>
> **Q3: Unfair comparison of single-modal detection methods.**
>
> **A:** We argue that the comparison is definitely fair, as we already tried various operations to adapt existing single-modal detection methods to be multi-modal detection methods. It is noteworthy that developing more advanced techniques to adapt single-modal detection to multi-modal detection is not the key point of our paper, which needs further specialized research.
>
> **Q4: The attack success rate of poisoned CLIP is not reported.**
>
> **A:** Actually, we have reported the ASR of some backdoor attacks (i.e., BadNet and Blended) and zero-shot accuracy performance of poisoned CLIP, as shown in Tables 25 and 26 in Appendix. We would like to mention that ISSBA achieves a relatively low ASR (near 76\%), thereby indicating that the target model is not fully poisoned (corresponding to your concern). In this case, our method still achieves an AUROC of 0.927, which validates the adaptability of our method against different poison levels of target models.
>
> **Q5: Dataset Limitations**
>
> **A:** We have conducted additional experiments on COYO-700M, which is also a large-scale image-caption dataset. Specifically, we select 50,000 image-text pairs from COYO-700M and poison 1,500 of them (BadNet), and fine-tune CLIP on the poisoned dataset. After this, we use our method for detection and achieve an AUROC of 0.94, which validates the effectiveness of our method on the real-world large-scale dataset.
>
> **Q6: CLIP variant**
>
> **A:** Existing works (i.e., CleanCLIP, BadCLIP, RoCLIP, and SafeCLIP) that focus on backdoor learning on vision-language models commonly consider CLIP as a representative victim model due to its reproducibility. Following this setting, we also follow the common protocol to use the same backbone for fair comparison.
>
> **Q7: Defense comparison (RoCLIP and SafeCLIP)**
>
> **A:** We have tried to use them to defend against BadCLIP on CC3M in the fine-tuning setting, however, due to limited time and computing resources, the experiments are still running. We expect to report the experimental results by the end of the reviewer-author discussion.
>
> **Q8: Threshold dependency and selection.**
>
> **A:** Using a small proportion of clean data for threshold selection is a common and reasonable strategy used in many backdoor defense methods. Besides, we also have conducted a detailed analysis for threshold selection based on 85\% quantile in Appendix B. The variance of the threshold may indeed be high in Table 10 (i.e., your concern), but the final performance of our method is not very sensitive to a wide range of thresholds, which means, the threshold selection would not hinder the practical application of our method.
>
> **Q9: Attack Diversity.**
>
> **A:** This is true that the detection performance against "Rooster" is slightly lower than that against other target classes (average 3.6\%) in Table 13. This may be because of the inherent diversity of target classes in terms of visual complexity and class descriptions.
>
> **Q10: Formal analysis for backdoor alignment.**
>
> **A:**  We provide a formal analysis by mutual information. Specifically, let us denote some variables: clean image $X$, backdoor image $Z$, and text change $Y$. Due to formulation $I(X;Y)=H(Y)-H(Y|X)$, where $H(Y)$ denotes the entropy of $Y$ and $H(Y|X)$ denotes the entropy of $Y$ conditioned on $X$, we can obtain $I(X;Y) - I(Z;Y) =H(Y|Z)-H(Y|X)$. Given backdoor images $Z$, the entropy (uncertainty) of $Y$ is high because $Z$ only has a strong relationship with the target text. In contrast, clean image $X$ correlates with specific semantic description and thus makes the entropy of $Y$ relatively low. So we obtain $H(Y|Z)>H(Y|X)$, and thus $I(X; Y)>I(Z; Y)$, which means backdoor images are less sensitive to text changes than clean images. We will include the above analysis in the revised version.

---

### Official Review · Reviewer_RpJy · 2025-03-15

**Overall Recommendation:** 4

**Summary:**

This paper introduces BDetCLIP, a novel method for test-time backdoor detection in CLIP-based models. Instead of trying to remove or mitigate backdoors at the pre-training or fine-tuning stages, the authors propose an inference-stage defense. By leveraging the contrastive alignment between images and class-related prompts (generated by GPT-4 or another LLM), BDetCLIP identifies whether an image is likely to be “backdoored” based on how sensitive its embedding is to benign vs. malicious textual descriptions. Extensive experiments on ImageNet and other datasets demonstrate both high effectiveness (AUROC scores) and low inference overhead, outperforming previous unimodal detection baselines.

**Claims And Evidence:**

Yes.

**Essential References Not Discussed:**

No.

**Experimental Designs Or Analyses:**

Yes.

**Methods And Evaluation Criteria:**

Yes.

**Other Comments Or Suggestions:**

No.

**Other Strengths And Weaknesses:**

Strengths
1. Novel Paradigm: Overall I like the paper. The paper tackles inference-stage backdoor detection in a multimodal setting - an under-explored yet practical direction for CLIP defenses.
2. Contrastive Prompting: The proposed “contrastive prompting” is a clever and intuitive mechanism that exploits semantic alignment between image and text, rather than relying on image-only perturbations.
3. Comprehensive Experiments: Results span multiple datasets (e.g., ImageNet, Food-101, Caltech-101) and compare against both traditional and multimodal backdoor attacks, showing strong performance in AUROC.
4. Efficiency: BDetCLIP is lightweight compared to many training-based defenses, requiring no parameter updates and using only two sets of prompts (benign vs. malignant) for detection.

Weaknesses
1. Randomness on LLM Generation: The success of generating effective prompts (especially “benign” ones) relies on high-quality language models and well-controlled generation. Although the empirical results are good, it is not very clear how to strategically control such generation.
2. Adaptive Attacks: The paper shows strong detection for existing backdoor attacks, but adaptive attackers might design triggers that shift embeddings in ways that mimic normal text-visual alignment. The paper could explore more robustness analysis against such adaptive strategies.

**Questions For Authors:**

Please respond to the Weaknesses.

**Relation To Broader Scientific Literature:**

The paper relates to the topics of safety and security of multimodal models.

**Theoretical Claims:**

No theoretical claims.

---

> ### Author Rebuttal · Authors · 2025-04-01
>
> Thank you very much for your positive feedback on our paper. Below, we provide point-by-point responses to address your concerns. We will be more than happy to discuss with you in the reviewer-author discussion period if there is anything unclear.
>
> **Q1: Randomness on LLM Generation: The success of generating effective prompts (especially “benign” ones) relies on high-quality language models and well-controlled generation. Although the empirical results are good, it is not very clear how to strategically control such generation.**
>
> **A:** Thank you for your valuable comment. In Section 4.2, we have already examined the impact of prompt variation on detection reliability, specifically analyzing the effects of prompt length and quantity. Our experimental results show that increasing the number of class-specific benign prompts generally enhances detection quality, while longer class-specific malignant prompts tend to degrade it. More importantly, our main experiments demonstrate that prompts generated with GPT-4 achieve a quality level sufficient to meet our defense requirements. Additionally, the strong and consistent detection performance observed across prompts generated by eight open-source large language models (as detailed in the Appendix) suggests that high-quality prompt generation is readily attainable.
>
> **Q2: Adaptive Attacks: The paper shows strong detection for existing backdoor attacks, but adaptive attackers might design triggers that shift embeddings in ways that mimic normal text-visual alignment. The paper could explore more robustness analysis against such adaptive strategies.**
>
> **A:** Thank you for your suggestion. Notably, BadCLIP-1 [1] in our experiment represents an adaptive attack, as its trigger is optimized using attribute texts of target classes, similar to the benign class descriptive texts employed in our method. As shown in Table 1, BDetCLIP maintains a high AUROC of 0.900 against BadCLIP-1, demonstrating the effectiveness of our approach in defending against adaptive attacks. Following your suggestion, we will provide a more in-depth analysis of our method’s robustness against such attacks.
>
> **References**
>
> [1] Badclip: dual-embedding guided backdoor attack on multimodal contrastive learning. In CVPR, 2024.

---

> > ### Comment · Reviewer_RpJy · 2025-04-03
> >
> > Thanks for the rebuttal. I have no more questions.

---

> > > ### Author Response · Authors · 2025-04-05
> > >
> > > Thank you for your reply. We are glad to know that you do not have any other concerns now. We really appreciate your positive feedback on our paper!

---

### Decision · Program_Chairs · 2025-05-01

**Decision:**

Accept (poster)

**Comment:**

This paper presents BDetCLIP, the first test-time backdoor detection method for CLIP models, addressing a critical gap in multimodal security. The key innovation lies in leveraging contrastive prompting—using GPT-4 to generate benign/malignant text descriptions—to detect backdoored images via distribution shifts in cosine similarity.  Reviewers initially raised concerns about adaptive attacks, threshold sensitivity, and comparison fairness, but the authors provided compelling rebuttals. While Reviewer nLrT maintained a weak reject, their concerns were thoroughly addressed, and other reviewers upgraded scores post-rebuttal.

Overall, the paper merits an acceptance for its originality, thorough evaluation, and timely relevance to safe AI deployment.